# On the Benefits of Instance Decomposition in Video Prediction Models

## Abstract

Video prediction is a crucial task for intelligent agents such as robots and autonomous vehicles, since it enables them to anticipate and act early on time-critical incidents. State-of-the-art video prediction methods typically model the dynamics of a scene jointly and implicitly, without any explicit decomposition into separate objects. This is challenging and potentially sub-optimal, as every object in a dynamic scene has their own pattern of movement, typically somewhat independent of others. In this paper, we investigate the benefit of explicitly modeling the objects in a dynamic scene separately within the context of latent-transformer video prediction models. We conduct detailed and carefully-controlled experiments on both synthetic and real-world datasets; our results show that decomposing a dynamic scene leads to higher quality predictions compared with models of a similar capacity that lack such decomposition.

## 1 Introduction

Video prediction is the task of predicting future frames based on past frames; it has many applications including autonomous driving (Yang et al., 2024), weather forecasting from satellite images (Ravuri et al., 2021), and even building general world models (Wang et al., 2024). Predicting future frames is challenging, since images are high-dimensional and result from the combination of multiple objects' appearances, dynamics and mutual interactions. For example, consider the environment observed while driving a car. How this scene will develop in the immediate future is dependent on all elements in the scene (e.g., cars, pedestrians, dogs) and their individual pattern of movement, including complex interactions with both static and moving parts of the scene (e.g., a car stopping at a traffic light or a dog following its owner on a leash). Hence, the complexity of the frame prediction task rises quickly as more objects with different motions interact in a scene, and with this, the size and training data required by prediction models. In this article we experimentally investigate the hypothesis that modelling explicitly the motion of the main objects in a scene and their interaction allows for better video prediction without need for larger models or additional training data.

To handle this complexity, one solution is to decompose the scene into parts (Sun et al., 2023; Bei et al., 2021; Lee et al., 2021; Hsieh et al., 2018). This enables modeling the appearance and dynamics of each part separately during prediction, thus reducing computational cost and increasing statistical efficiency. Several works have achieved promising results by such approaches, using different choices of decomposition. For example, Hsieh et al. (2018) uses DRNet (Denton et al., 2017) to learn a disentangled representation of appearance and 2D pose, while Bei et al. (2021); Lee et al. (2021) use semantic segmentation models, and Sun et al. (2023) separates the foreground, motion and background. Wu et al. (2023) uses object-centric representation learning (Locatello et al., 2020) to separate objects without supervision, and model the dynamics with a multi-slot transformer.

While those approaches achieve impressive results, they do not focus on measuring the benefit of object decomposition in a scientifically-controlled way, i.e., keeping confounding factors such as the number of network parameters, architecture or latent dimensionality constant. Moreover, these works (Gao et al., 2022; Wang et al., 2022; 2018) did not use the modern large latent-space transformer architectures that now

yield excellent results on diverse domains of videos (Yan et al., 2021; Wu et al., 2024); they instead used older, smaller CNN- or RNN-based models.

In this work, we perform a detailed study of the benefits of explicit modeling of separate objects' motions during video prediction, using modern latent transformer models. Rather than introducing an entirely new model, we develop a family of architectures similar to VideoGPT, MOSO and slotformer (Yan et al., 2021; Sun et al., 2023; Wu et al., 2023), that supports both single-slot (i.e., jointly modeling the whole scene) and multi-slot (i.e., per-object) representations in a unified framework. This allows us to perform controlled experiments on the benefits of object decomposition and on strategies for modeling interactions. Specifically, we adopt a hierarchical approach that explicitly decomposes a dynamic scene into individual objects using an instance segmentation model, before encoding these into separate latent spaces. Because objects of the same class will have similar motion patterns, for example different cars or different pedestrians, it is not efficient to model each object's dynamics by a separate slot. Therefore, we mitigate the inefficiency of having separate network parameters per object instance (Villar-Corrales et al., 2023) by sharing parameters across all instances of each class.

We find that, even with large transformers, object decomposition leads to considerable improvements in handling complex scenes with multiple interacting objects compared to non-object-centric predictors with similar parameter counts and latent dimensions.

Our main contributions are as follows:

- We present the first systematic and comprehensive analysis of the benefits of explicit object decomposition for latent transformer video prediction models.

- To achieve this, we develop a scalable framework for video prediction that supports both the single- and multi-slot settings.

- We mitigate statistical inefficiencies in object-centric video predictors by sharing weights (and thus knowledge about object dynamics) across slots within each object class.

## 2 Related Work

### 2.1 Recurrent models for video prediction

Early video prediction models were typically based on the combination of Convolutional Neural Networks (Krizhevsky et al., 2012) and Recurrent Neural Networks, often LSTMs (Shi et al., 2015; Wang et al., 2022; 2018; Chang et al., 2022; Gao et al., 2022; Denton & Fergus, 2018). Lee et al. (2021) proposed a method to predict future semantic maps, then used those predicted maps to warp the actual future frames from the past RGB frame. Bei et al. (2021) proposed a similar approach, decomposing the scene with a semantic map, and using separate pathways to model the dynamics of different semantic classes. Of these, some methods are deterministic, i.e., make a single most-likely prediction of the future (Shi et al., 2015; Wang et al., 2018), while others are stochastic, i.e., sample an autoregressive posterior distribution on possible future frames (Denton & Fergus, 2018; Lee et al., 2021). We focus on the stochastic setting in this work since the deterministic models tend to predict and converge to the mean of the possible future, as well as typically producing sharper predictions (Ohayon et al., 2023).

### 2.2 Transformer models for video prediction

Following their success on text (Vaswani et al., 2017) and images (Dosovitskiy et al., 2021), Transformers have also been applied to video prediction. A common approach is to first use an encoder network to map the original video frames into a sequence of lower-dimensional latent vectors. Most models use VQ-VAE (van den Oord et al., 2017) or VQ-GAN (Esser et al., 2021) as their encoding network due to their high fidelity reconstruction of original frames, and discrete latent space that enables treating the latents similarly to text tokens. Yan et al. (2021) proposed the first autoregressive video prediction model based on VQ-GAN and a decoder transformer to predict future frames; iVideoGPT Wu et al. (2024) improves performance

further. Gupta et al. (2022) proposed a similar method that uses VQ-VAE and transformer, but trains with iterative masking to let it gradually capture the motion patterns in a video. Sun et al. (2023) proposed a pipeline that decomposes the dynamic scene into motion, object and background, then uses a stochastic transformer to predict future frames in latent space. Our work also uses a latent transformer, but with an explicit decomposition of the latent space into separate objects, and cross-attention to capture object interactions.

### 2.3 Diffusion models for video prediction

The invention of diffusion models (Sohl-Dickstein et al., 2015; Ho et al., 2020) and the computationally faster latent diffusion (Rombach et al., 2022) brought significant improvement on many generative tasks. Latent diffusion was originally designed to generate high-resolution images, but has now been applied to video (Blattmann et al., 2023b;a; Brooks et al., 2024). Ho et al. (2022) use a diffusion model to generate long videos via a joint training paradigm with conditional sampling. Höppe et al. (2022) use a slightly different training process that instead of adding noise to the entire video, randomly keeps some of the input frames without noise. Yu et al. (2023) proposed an interesting way of modeling latent vectors in three directions by slicing 3D feature vectors along different axes. SORA (Brooks et al., 2024) alongside with Veo3 (DeepMind, 2025) is the state-of-the-art video generation model, and can generate extremely realistic videos by using diffusion with a transformer architecture. It is able to accurately generate complex interactions involving multiple objects (Liu et al., 2024). However, in order to train these kind of models, it is extremely expensive in terms of data and computation power.

### 2.4 Object-centric video prediction

Object-centric representation learning aims to learn decomposed representations of images (Locatello et al., 2020; Engelcke et al., 2020) or videos (Jiang et al., 2019; Zhou et al., 2022) without supervision. This can be used to aid video prediction by learning an object-centric predictor (typically a transformer) over the resulting representations (Kipf et al., 2022; Li et al., 2021; Sajjadi et al., 2022; Singh et al., 2022). Villar-Corrales et al. (2023) use an attention mechanism to learn the relationship between different objects in the video sequence and achieved good results on synthetic CLEVRER Yi* et al. (2020) dataset. Schmeckpeper et al. (2021) use Mask R-CNN (He et al., 2017) to get bounding boxes for each entity in the scene, then predict the next state of each bounding box from a single frame. Finally, Henderson & Lampert (2020); Henderson et al. (2021) proposed self-supervised object-centric approaches that predict frames via latent 3D objects and scene structure from 2D video. Instead of learning the object centric information from the raw video frames, we use segmentation masks to decompose the objects by using a pre-train segmentation model.

### 2.5 Cross-attention

Our model uses cross-attention between instances to capture object interactions. Similar ideas have been used in many other domains, e.g., (Zhu et al., 2022) use pairwise cross-attention to re-identify pedestrians; Shi et al. (2025) use cross-attention to fuse information from audio and video for emotion recognition; Lee et al. (2023) use pairwise cross attention on video action recognition; Rombach et al. (2022) uses cross attention between image features and text embeddings for conditional image generation. In this work, we use cross-attention to model the potential interaction between each object, and also evaluate the impact of using cross-attention to handle object interactions in a dynamic scene.

## 3 Methodology

Let $X^{1:T} = \langle x^1, x^2, ..., x^T \rangle$, be a sequence of $T$ RGB frames from a video clip, where $x^t \in \mathbb{R}^{h \times w \times 3}$. Our goal is to learn a probability distribution on $M$ future frames $X^{T+1:T+M}$, conditioned on the $T$ preceding frames $X^{1:T}$.

We hypothesize that predicting future frames is more effective when modeling each object or instance separately rather than modeling the entire scene at once. Moreover, when objects are decomposed, we aim

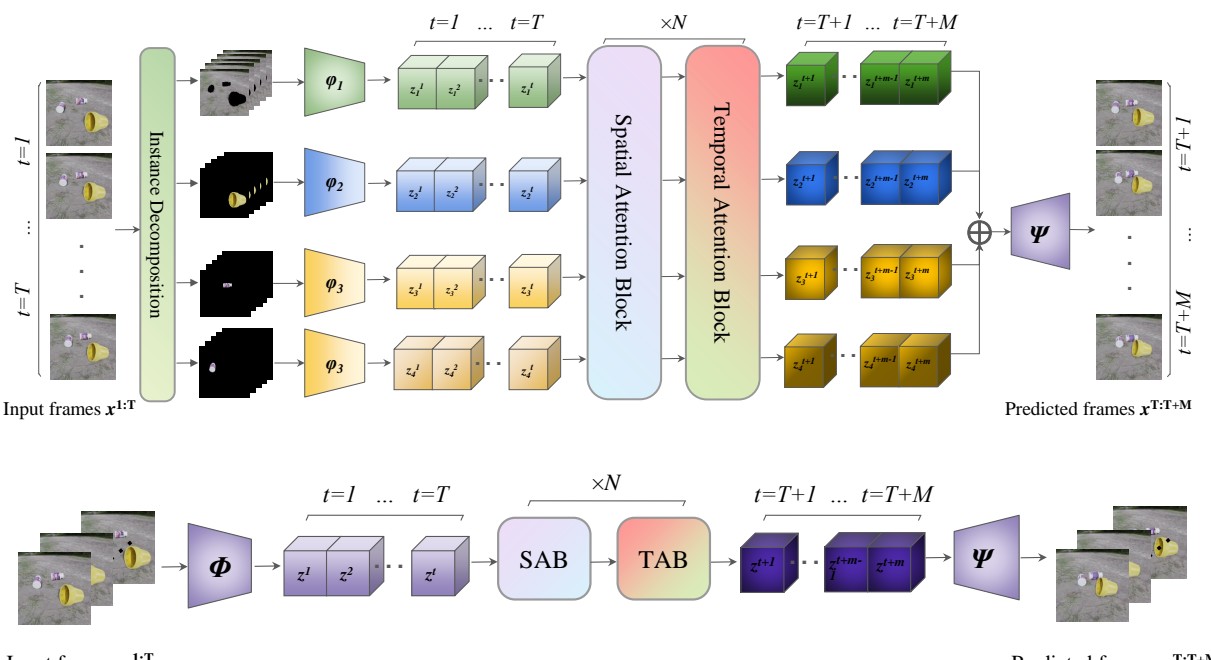

Figure 1: **Top:** Our proposed multi-object interacting model **SCAT**. First, the input frames are decomposed via a segmentation model, then each decomposed sequence passes through class-specific encoder to convert the 2D frames into latent representations; then, class-specific transformer blocks learn and predict the dynamics of each instance and its relationships with other instances in latent space; lastly, the predicted latent representation are decoded via joint decoder to reconstruct the predicted RGB frames. **Bottom:** The non-decomposed single-slot variant **SiS** where the scene is modeled globally and jointly. In **SNCAT**, the cross-attention module is replaced with same-capacity feedforward network, please see Figure 2 for more details.

to measure the degree to which cross-attention enables learning interactions among objects, thus making prediction more accurate.

To test this hypothesis, we design a family of models that support differing degrees of object decomposition and interaction within a unified framework. We decompose a scene into individual objects using instance segmentation models (Reis et al., 2023; Lüddecke & Ecker, 2022). The video prediction models then comprise an *object-aware auto-encoder* (OAAE) (Section 3.1), which extracts latent representations for each object, and a multi-object transformer (Section 3.2) that predicts future latent representations conditioned on previous ones; the OAAE is used to decode these future latents back into video frames. To test our hypotheses, we propose three variants of our overall pipeline:

- **Single Slot (SiS)**: Objects are not modeled separately; frames are encoded with a single encoder, and a standard (not object-centric) transformer network is used to predict future frames; this is similar to VideoGPT (Yan et al., 2021).

- **Stochastic non-Class Attended Transformer (SNCAT)**: The scene is decomposed into instances; both the encoder and predictor have one slot for each object in the scene, with parameters shared across instances of the same class, but no interactions among different object slots in the transformer.

- **Stochastic Class Attended Transformer (SCAT)**: Our full model, which encodes instances separately, then uses a multi-slot transformer for future prediction, with cross-attention to capture object interactions.

The overall pipeline of the fully-interacting decomposed **SCAT** and single slot **SiS** models is shown in Figure 1.

### 3.1  Object-aware autoencoder

We now discuss the encoder we use for extracting the latent representation of a video, which will be used in Section 3.2 as a lower-dimensional space for future prediction. We first explain the object-aware autoencoder (OAAE) as used in the **SCAT** and **SNCAT** models, then give a brief explanation of the simpler (non-object-centric) variant used in **SiS**.

**Instance decomposition.**   Let $x \in \mathbb{R}^{h \times w \times 3}$ be a frame in an RGB video sequence of width $w$ and height $h$. It is decomposed into a set of $N$ instances with corresponding class labels using an off-the-shelf segmentation model (Reis et al., 2023; Lüddecke & Ecker, 2022). The segmentation returns $N$ non-overlapping binary masks, each belonging to one of $m$ object classes $c_k \in \{1, \ldots, m\}$; we then multiply the input frame by the respective masks to isolate each object. The $k^{\text{th}}$ masked instance is denoted by $\tilde{x}_k$ for $k \in \{1, 2, \ldots, N\}$, and its class is denoted as $c_k$. Assuming the segmentation is panoptic and covers all pixels of the frame, the original frame can be reconstructed by recombining all instances of all classes additively as follows:

$$x = \sum_{k=1}^{N} \tilde{x}_k \tag{1}$$

**Instance embedding.**   We modify the standard VQ-VAE (van den Oord et al., 2017) model to have a set of encoders $\Phi = \{\phi_1, \phi_2, ..., \phi_m\}$ and a set of embedding code books $E = \{e_1, e_2, ..., e_m\}$, each associated with an individual semantic class. Each instance frame $\tilde{x}_k$ is passed to the corresponding encoder $\phi_{c_k}$ and quantized with $e_{c_k}$ to produce a latent vector $\tilde{z}_k$:

$$\tilde{z}_k = e_{c_k}^i \text{ where } i = \arg\min_j (\|\phi_{c_k}(\tilde{x}_k) - e_{c_k}^j\|_2) \tag{2}$$

The quantized representations are then concatenated into a single vector $z = \bigoplus_{k=1}^{N} \tilde{z}_k$ that encodes the complete frame $x$ (where $\bigoplus$ denotes concatenation operation).

For convenience, we will use the notation $z = \Phi(x)$ to denote the overall encoding operation. This latent representation $z$ can then be passed to a single joint decoder $\Psi$ to reconstruct the full frame, i.e., $\hat{x} = \Psi(z)$. After each up-sampling convolutional layer in the decoder, we incorporate Frequency Complement Modules (FCM) (Lin et al., 2023) to learn not only from the target frame but also from feature maps between encoder and decoder.

**Loss function.**   Since our OAAE is a multi-object extended version of the original VQ-VAE (van den Oord et al., 2017) with some features of FA-VAE (Lin et al., 2023), we also extend the original loss functions correspondingly. There are 4 losses: feature loss, commitment loss, vector quantisation loss (VQ loss) and reconstruction loss. Following (Lin et al., 2023), we impose a loss on feature maps, not only on the final pixels; similar to them we use focal frequency loss (FFL (Jiang et al., 2021)) between the output of encoder convolution layers and decoder FCM layers:

$$\mathcal{L}_{feature} = \sum_{c=1}^{m} \sum_{l=0}^{L-1} FFL(f_l^c, g_{L-l}) \tag{3}$$

where $c$ indexes encoders (recall there is one per class), $l$ indexes over convolutional layers in the $c^{\text{th}}$ encoder and $L - l$ over corresponding FCM layers in the decoder ($L$ is the total number of decoder layers), $f_l$ represents the feature map of the $l^{\text{th}}$ encoder layer, and $g_l$ that of the $l^{\text{th}}$ FCM module in the decoder. The VQ and commitment losses are similar to the original VQ-VAE, except we compute these for each class $c$ and instance $k$, then sum over these:

$$\mathcal{L}_{VQ} = \sum_{k=1}^{N} \|sg[\phi_{c_k}(\tilde{x}_k)] - e_{c_k}\|^2 \tag{4}$$

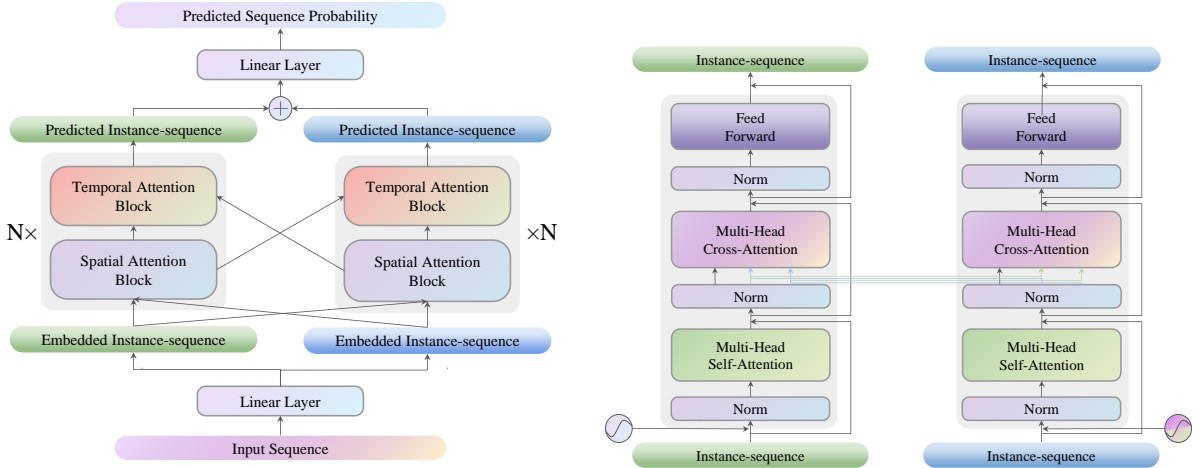

Figure 2: **Left:** Architecture of the multi-object latent transformer. **Right:** Detail of spatial and temporal attention blocks. **SCAT**'s block structure is as illustrated, **SNCAT** removes the pathways of cross attention in both sides of the diagram, and in the right diagram, Multi-Head Cross attention is replaced with same-capacity feed forward block.

$$\mathcal{L}_{commitment} = \sum_{k=1}^{N} \| \phi_{c_k}(\tilde{x}_k) - \text{sg}[e_{c_k}] \|_2^2 \tag{5}$$

where sg is the stop-gradient operator. Finally, the reconstruction loss is composed of pixel-space and frequency-space terms calculated between the reconstructed and original frames:

$$\mathcal{L}_{recon} = -\log p(x|\Psi(\Phi(x))) + FFL(x, \Psi(\Phi(x))) \tag{6}$$

Putting all four terms together yields the final loss function for training OAAE:

$$\mathcal{L}_{oaae} = \mathcal{L}_{recon} + \alpha\mathcal{L}_{feature} + \mathcal{L}_{VQ} + \beta\mathcal{L}_{commitment} \tag{7}$$

where $\alpha$ and $\beta$ weight the different loss terms. Once the OAAE is trained, we denote the latent representation for the frame $x^t$ at time step $t$ as $z^t$. This provides a structured and disentangled representation, capturing $N$ instances across $m$ classes.

**Variations of the OAAE.** In order to measure whether object decomposition helps with prediction, we also define a non-decomposed version of the VQ-VAE, for use in model **SiS**. This only takes the original non-segmented frame as input. It is processed by a single encoder, with the latent size matched to the total latent size (over all instances) for model **SCAT**. In terms of losses, $L_{recon}$ remains unchanged, $L_{VQ}$, $L_{commitment}$ and $L_{feature}$ will be a modified to a single term without summation since there is now a single encoder and codebook, and feature maps from just one instance (e.g., the whole frame). For the **SNCAT** model variant, the OAAE is identical to the main version for **SCAT**, only the subsequent transformer stage is different.

### 3.2 Prediction Model

Using the OAAE, a video clip $X$ is encoded as a sequence of latent representations $Z = \langle z^1, z^2, \dots, z^T \rangle$. To learn the instance dynamics and its relationship with other instances, we modify the original decoder-only transformer (Vaswani et al., 2017; Radford et al., 2018) into a slot-per-instance auto-regressive transformer that has cross-attention between instances, and shares parameters across instances of each class.

Our transformer consists of alternating attention and feed-forward blocks. However, unlike typical 1D transformers, it includes factored spatial and temporal attention blocks; each of these is applied both for

self-attention (i.e., each instance independently attending to other locations / time-points of itself), and cross-attention (i.e., each instance attending to different locations / time-points of all other instances). We use PreNorm (Xiong et al., 2020) in each transformer block. The output vectors for each instance from the last transformer layer are concatenated and passed through a linear layer. The output size matches the number of embeddings in OAAE, allowing the model to predict the probability of possible indices of future frames.

Because the latent vectors produced by the OAAE are a concatenation of each object instance's latent encoding, we can write the sequence of latent encodings in the video for each individual object instance as $\tilde{Z}_k = \langle z_k^1, z_k^2, ..., z_k^T \rangle$ where $k$ denotes the $k_{th}$ instance.

**Spatial and temporal extensions of attention layers.** Since an instance latent sequence $\tilde{Z}_k$ has a 3-dimensional shape $t \times (h \times w) \times c$, where $c$ represents embedding dimension in OAAE, it encompasses both temporal and spatial information. Merely flattening the latent vector to form the video sequence in latent space risks losing crucial spatial details. Hence, inspired by (Sun et al., 2023), all attention layers are applied in both spatial ($h \times w$) and temporal $t$ dimensions. This ensures the model can capture not only the temporal relationships within the sequence but also the important spatial information embedded within each latent representation.

**Instance-level self-attention.** For each latent instance frame $z_k^t$ in the sequence, we first apply learnable positional embeddings. This embedding is added to the input features prior to self-attention to provide the model with information about the position of each instance within the sequence. Scaled self-attention is then applied to each instance sequence separately in order to learn instance-specific dynamics:

$$\text{SA}_c(\tilde{Z}_k) = \text{softmax}\left(\frac{Q_k K_k^T}{\sqrt{d_k}}\right) V_k \tag{8}$$

where SA denotes instance-specific self-attention for objects of class $c$, $Q_k, K_k^T$, which $T$ denotes transpose, and $V_k$ are the key, query and value calculated by a linear function on $\tilde{Z}_k$; $\frac{1}{\sqrt{d_k}}$ is a scaling factor that prevents excessively large values in the attention score. Following self-attention, we apply a further linear projection layer.

**Instance-level cross-attention.** After the self-attention layer that treats each instance separately, we apply cross-attention between instances to learn the potential relationships and interactions between objects. In this layer, each instance attend the space/time dimensions of each of the other instances:

$$\text{CA}(\tilde{Z}_k) = \bigoplus_{i=1...N,\, i \neq k} \text{softmax}\left(\frac{Q_k K_i^T}{\sqrt{d_k}}\right) V_i \tag{9}$$

Here CA denotes the cross-attention operation between instance $k$ and the remaining instances. The value $V_i$ and key $K_i$ are derived from $\tilde{Z}_i$, while the query originates from $\tilde{Z}_k$. The cross-attention layer's output, being $n-1$ times larger than the input because of concatenation, is reduced to the original size through a linear layer.

**Training and inference.** The model outputs probabilities over the codebook indices from OAAE, and we use cross-entropy loss to minimize the difference between the predicted and actual distributions. During training, all model variants are trained with teacher forcing on 10-frame clips. Before the forward pass, 10% noise sampled from a standard normal distribution $\mathcal{N}(0,1)$ is added to the input frames. During inference, autoregressive sampling is used, starting from an initial sequence of conditioning frames, with softmax temperature treated as a hyperparameter.

**Variants of the transformer.** We have described the transformer as used in the full model **SCAT**. In the non-interacting model **SNCAT**, cross-attention is simply replaced by a per-object feed-forward network of similar capacity. The single-slot version **SiS** has a single, larger latent vector for the whole scene instead of separate latents for each object, and we also increase the hidden dimensionality of the transformer (in fact

resulting in considerably more parameters). The number of feed-forward and self-attention layers remains the same.

## 4 Experiments

We perform a series of experiments to measure the benefit of separately modeling the dynamics of objects during video prediction. Our focus is on comparing different model variants in a controlled setting, keeping model capacity approximately equal but changing whether the latent representation is decomposed over objects, and whether interactions between objects are modelled if so. In addition, to place our results in context, we perform a comparative evaluation against other recent video prediction models under similar conditions.

### 4.1 Experimental protocol

Each model is given five frames as input, then predicts the following 5–25 frames depending on the dataset. We use $64 \times 64$ resolution for all datasets; further details on hyperparameters are in the appendix. The models (variants of both encoder and predictor) are implemented in PyTorch and trained from scratch on a single NVIDIA RTX 3090 GPU, reflecting our emphasis on computational efficiency and model scalability; further implementation details are given in the appendix. To ensure a rigorous comparison that focuses on the benefit of instance decomposition, we ensure the numbers of parameters in each model are as similar as possible. Our focus is not on achieving state-of-the-art performance but rather on analyzing the benefits of explicit object-centric modeling within a balanced and controlled setting. For quantitative evaluation, we report Peak Signal-to-Noise Ratio (PSNR) (Horé & Ziou, 2010), Structural Similarity (SSIM) (Wang et al., 2004), and Learned Perceptual Image Patch Similarity (LPIPS) (Zhang et al., 2018). PSNR measures the pixel-wise fidelity between images; SSIM evaluates perceptual similarity in terms of luminance, contrast, and structure; LPIPS uses deep network features to capture perceptual similarity. We focus on LPIPS scores in this paper, because it is more aligned with human perception while PSNR and SSIM are sensitive on slight misalignment that leads to poor scores. The results are obtained by sampling with 10 different temperature values ranging from 0.1 to 1.0 (from low to high stochasticity), using softmax to sample likely future indices—yielding 11 evaluations in total. For each test video sequence, 25 samples are generated for the same input, which is standard in stochastic prediction tasks (Denton & Fergus, 2018), and the best one (in terms of metric score) is selected. After evaluating each model on each dataset, bootstrapping is used to estimate the spread. We sampled 10000 same-sized evaluation sets with replacement then calculated the mean and standard deviation of these sets, which are reported in the tables and figures.

### 4.2 Datasets

We conduct experiments on five different datasets characterized by weak and strong interactions. We define weak interactions as scenarios where the dynamics of an instance are unaffected by other instances, or minimally so. In contrast, strong interactions involve instances significantly affecting each other's dynamics, such as during collisions. Since our focus is measuring how the interaction between objects are handled by explicit decomposition and cross-attention, we do not address the problem of background motion in this paper, therefore none of the dataset we use features moving background.

The first weak interaction dataset we use is the **KTH** human action dataset (Schuldt et al., 2004). This includes six action types performed by 25 individuals. Although the primary focus is on the person, there remains some slight interaction between the person and the background, such as shadows cast by the individual on the background. Following MOSO (Sun et al., 2023), we use videos of persons 1-16 for training and 17-25 for testing. We used (Lüddecke & Ecker, 2022) to segment the person and the background. Each model is given five frames and required to predict 15 future frames.

The second weak interaction dataset is the **Real-Traffic** dataset from Ehrhardt et al. (2020). This comprises video clips taken from a CCTV camera overlooking a highway intersection. The background is static, and only the cars are moving in the scene; there are up to five cars per clip. The original dataset contains 615 video clips with various lengths, we split the dataset into a more standardized 10 frames per clip with 5,089

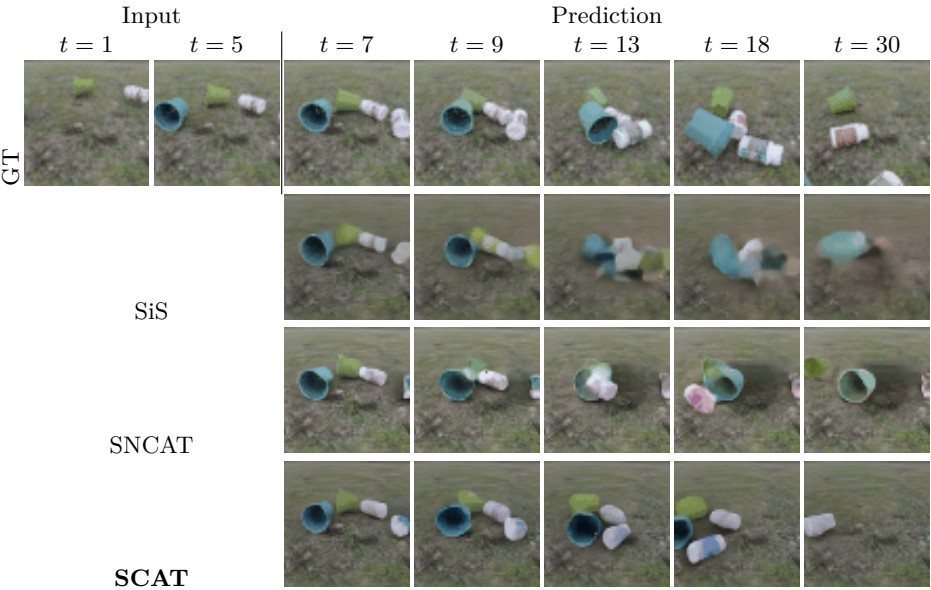

Figure 3: Comparison of different model variants on the **Kubric-Real** dataset. SCAT successfully predicted that the blue pot bounced away whereas SNCAT neglected the interaction between other objects and let the blue pot go through from other objects. The single-slot model SiS fails to capture the appearances well, yielding indistinct predictions for later frames.

clips for training and 2,181 for validation. During inference, the models are given five frames and required to predict five future frames. We used YOLOv8 (Reis et al., 2023) to extract each instance. Each car's motion is independent of other cars most of the time; however, interactions do occur, such as when a car stops before the intersection, causing other cars behind it to slow down. For quantitative evaluation, we therefore identify a subset of video clips from the test set with the strongest interactions. We calculate the distances between centroids of different cars, and select clips where the distance between any pair of cars is less than 25% of the image size; this yields a test set of 807 clips.

For strong interactions, we used Kubric (Greff et al., 2022) to generate a series of synthetic datasets inspired by CLEVRER (Yi* et al., 2020) but exhibiting stronger interactions and more visual complexity. Full details on the dataset generation (and corresponding code) are included in the appendix. Specifically, **CLEVR-2** contains scenes with two spheres with random velocity sampled such that they will collide; **CLEVR-3** scenes are similar but include another sphere that does not interact with the first two. **Kubric-Real** uses a realistic background and replaces the basic geometric objects with 3D-scanned objects—bottles and pots, since these exhibit interesting dynamics due to their cylindrical shapes.

### 4.3 Internal and External Evaluation

As we mentioned previously in the experimental protocol, that the best performing sample is selected from 25 samples. Here we first analyze how this selection process will impact the prediction performance. Both qualitative and quantitative results indicates the best-case predictions are closer to the ground truth compared to the average or worst cases. As illustrated in Figure 4 and Figure 9, the predicted object trajectories in the best-case sample are more accurately aligned with the ground truth, whereas in the worst-case sample, the object positions deviate substantially. However, this does not mean the prediction quality is poor, but indicates that the object trajectories are not close to the ground truth which is expected in stochastic sampling.

We now compare our model variants, to evaluate the benefit of explicit object-centric modeling in a controlled setting. Table 1 shows quantitative results on the two weak-interaction datasets. For **KTH**, the models are given five frames and required to predict 15 frames and for **Real-Traffic**, they are required to predict five

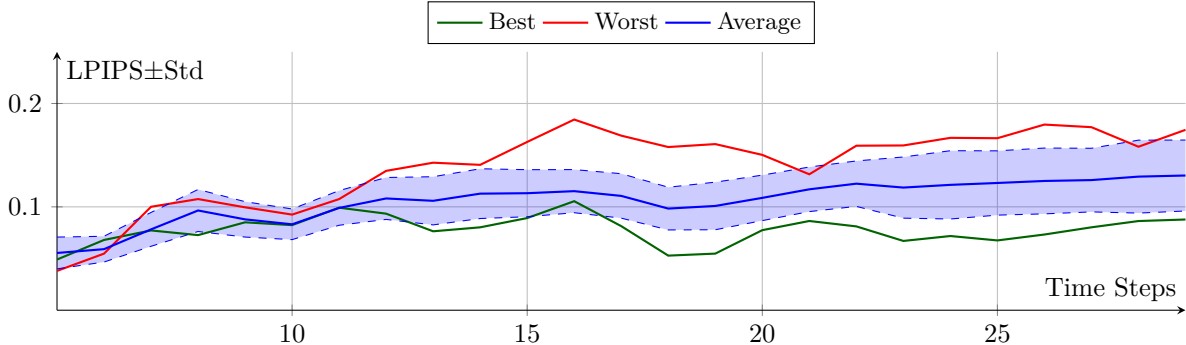

Figure 4: LPIPS metrics of worst, average and best cases of the sample shown in Figure 9; Note that the Standard deviation presented in this figure is obtained without using bootstrapping technique

frames. In both datasets the SCAT model performs better than the two other variants (SNCAT & SiS). First, modeling the scene separately by segmenting it at the instance level (SNCAT) leads to predictions comparable to modeling the whole scene at once (Single-slot model), while using a much smaller model (25M vs. 48M parameters on **KTH**, 27M vs. 286M parameters on **Real-Traffic**). In **KTH**, we see negligible decrease compared to SiS model, whereas in **Real-Traffic** a slight improvement has been made due to this dataset having more instances and stronger interaction between instance compared to **KTH**. Second, adding cross-attention to the model to handle potential interactions between instances (SCAT) leads to an improvement in performance across all metrics. Since **KTH** features a single instance with negligible interaction, the performance improvement is subtle on each metric: SSIM (+0.003), PSNR (+0.05) and LPIPS (-0.03). On **Real-Traffic**, which has more instances and higher interactions, consistent improvements are observed in all metrics (PSNR: +0.78, SSIM: +0.01, LPIPS: -0.007). These results confirm the computational advantage of both the decomposition and cross-attention components of the approach. From Figure 5, we can see that in **Real-Traffic** dataset, improvements are also shown in every time step of the prediction. In **KTH** dataset, since the interaction level is negligible, the improvement is not obvious.

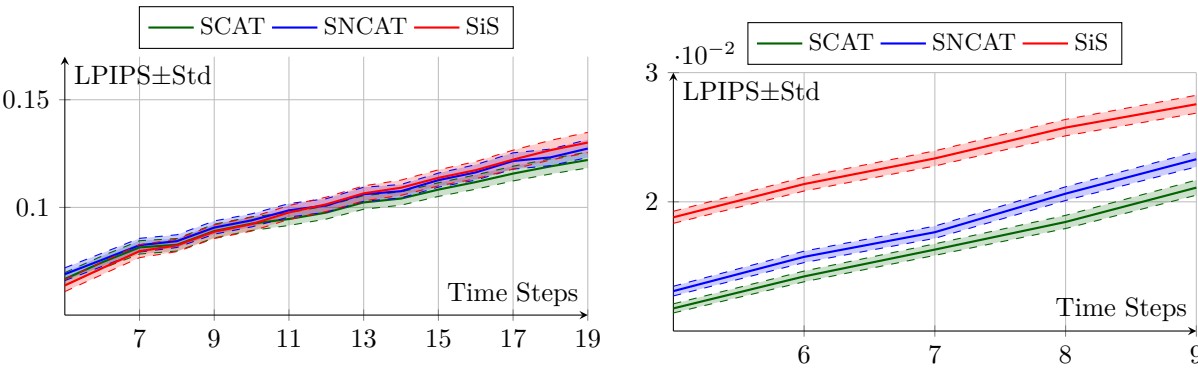

Figure 5: Mean and Std of LPIPS metric for **KTH(left)** and **Real-Traffic(right)** datasets

Table 1: Quantitative results on **KTH** and **Real-Traffic** datasets

| | KTH | | | | Real-Traffic | | | |
|---|---|---|---|---|---|---|---|---|
| | PSNR↑ | SSIM↑ | LPIPS↓ | Num-Prms | PSNR↑ | SSIM↑ | LPIPS↓ | Num-Prms |
| Single-Slot | 26.49±0.22 | 0.786±0.005 | 0.100±0.003 | 48M | 29.63±0.12 | 0.939±0.001 | 0.023±0.0005 | 286M |
| SNCAT | 26.36±0.17 | 0.785±0.005 | 0.101±0.003 | 25M | 30.02±0.12 | 0.946±0.001 | 0.018±0.0004 | 27M |
| SCAT | **26.54±0.18** | **0.789±0.004** | **0.097±0.003** | 23M | **30.41±0.12** | **0.949±0.001** | **0.016±0.0004** | 28M |

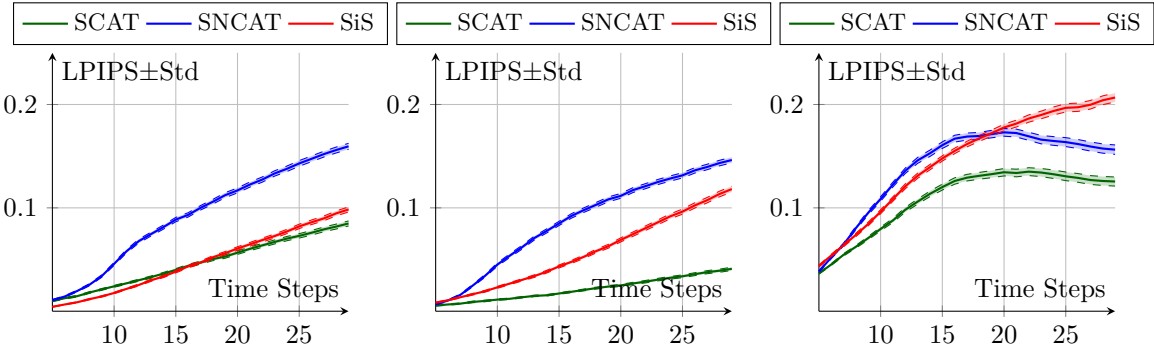

Figure 6: Mean and Std of LPIPS metric for **CLEVR-2(left)**, **CLEVR-3(middle)** and **Kubric-Real(right)** datasets

Table 2: Quantitative results on **CLEVR-2**, **CLEVR-3**, and **Kubric-Real** datasets

| | CLEVR-2 | | | | CLEVR-3 | | | | Kubric-Real | | | |
|---|---|---|---|---|---|---|---|---|---|---|---|---|
| | PSNR↑ | SSIM↑ | LPIPS↓ | Num-Prms | PSNR↑ | SSIM↑ | LPIPS↓ | Num-Prms | PSNR↑ | SSIM↑ | LPIPS↓ | Num-Prms |
| Single-Slot | **31.70±0.14** | **0.925±0.001** | 0.048±0.001 | 105M | 31.25±0.11 | 0.911±0.001 | 0.057±0.001 | 186M | 24.14±0.17 | 0.748±0.004 | 0.146±0.002 | 287M |
| SNCAT | 29.72±0.10 | 0.908±0.001 | 0.093±0.002 | 25M | 29.55±0.01 | 0.898±0.002 | 0.087±0.002 | 26M | 24.18±0.18 | 0.759±0.004 | 0.139±0.003 | 38M |
| SCAT | 31.11±0.12 | 0.919±0.001 | **0.047±0.001** | 25M | **34.42±0.14** | **0.947±0.001** | **0.022±0.001** | 26M | **25.13±0.19** | **0.789±0.004** | **0.108±0.003** | 40M |

Table 2 provides quantitative results on the strong-interaction datasets. On **CLEVR-2**, the SCAT model (PSNR: 31.11) performs similarly to the single-slot model (PSNR: 31.70) but outperforms it on LPIPS (0.047 vs. 0.048). In contrast, SNCAT performs worse than the single-slot model both on **CLEVR-2** and **CLEVR-3** datasets, this is due to the lack of cross-attention to model interactions between objects which lead to deformations of the spheres when collision happens. In **CLEVR-2**, where only two spheres colliding, SiS model can handle this simple interaction. However in **CLEVR-3**, where one sphere is added but not interacted with the original two, SiS model starts to struggle but SCAT performs best by a large margin. This also shows that SCAT's efficiency of modeling multiple objects' motion without the need of big sized model. In **Kubric-Real**, SNCAT preserves object shapes better than the single-slot model, which struggles with deformation after collision. SCAT outperforms both models in LPIPS (0.108 vs. 0.146 for the single-slot model) and SSIM (0.789 vs. 0.748 for the single-slot model), emphasizing the importance of cross-attention in more realistic and complex interaction scenes. Also, From Figure 6 we can see that due to the strong interactions, removing cross-attention makes SNCAT unable to beat the single slot model. In contrast, SCAT performed better than other two variants because of interaction handling with cross-attention. On **Kubric-Real**, note that towards the end of the prediction time frame, the prediction accuracy of **SCAT** and **SNCAT** starts to improve again. This is due to the fact that the moving object has either stopped moving or left the scene entirely. These results confirm our hypothesis that instance segmentation is important for video prediction and that cross-attention is an effective way to encode strong interactions. Moreover, without cross-attention, instance separation on its own is sufficient to achieve similar or better performance compared to the baseline single-slot model on complex scenes (**Real-traffic**, **Kubric-Real**) having more than two instances, with only a fraction of the parameters.

In addition to the model's prediction performance, we also measure the FLOPs of a single forward pass of our proposed variants to evaluate their computational efficiency, which shown in Figure 7. It shows both SCAT's and SNCAT's encoder FLOPs are slightly higher compared to SiS's encoder, this is expected because the variants with decomposition have individual encoder for an object class where single-slot encoder only have a single encoder. However, the total FLOPs of decomposed variants are smaller than the one without decomposition across different datasets even when the segmentation model is involved. This suggests the decomposed variants are more computationally efficient than non-decomposed variant.

Since SNCAT and SCAT variants depend entirely on the performance of instance segmentation model, we simulate under- and over-segmentation of an instance segmentation model with image processing techniques such as erosion and dilation. Figure 8 shows that with the increase of the kernel size, performance of

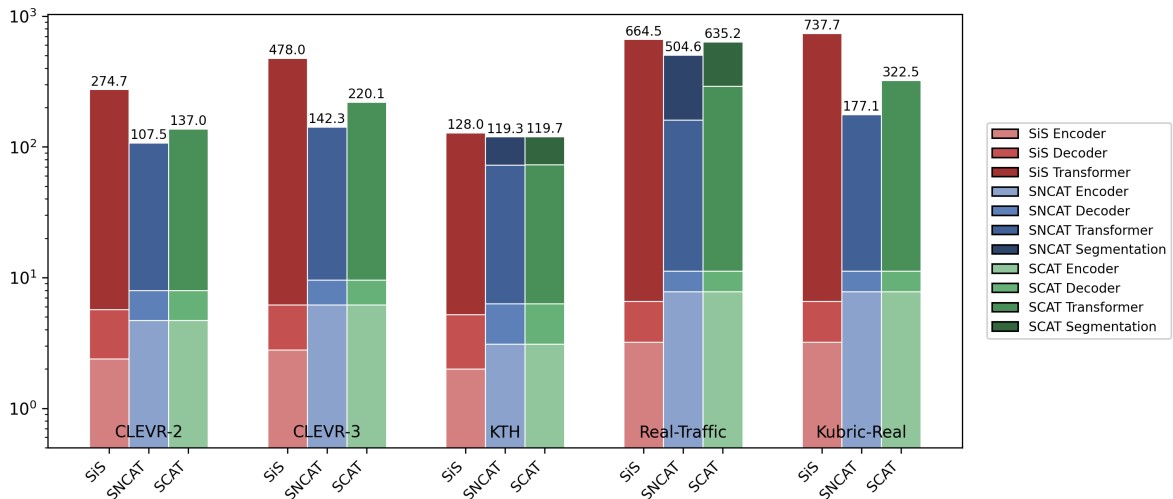

Figure 7: FLOPs (GMac) of a single forward pass comparison across different model variants. Note: Y-axis uses log-scale

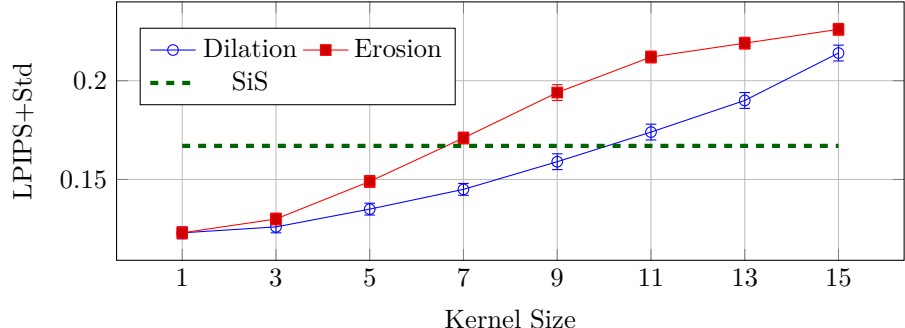

Figure 8: Impact of over- and under-segmentation on SCAT performance simulated via dilation and erosion operations on Kubric-Real dataset. We evaluated the samples generated by using **argmax** on logits to isolate the effect of dilation and erosion from stochasticity.

SCAT is decreased. This suggest that when the segmentation model's performance is poor the proposed pipeline's performance will also decrease accordingly. It is worth noting that although both over- and under-segmentation has negative impact on the prediction quality of SCAT, we can see when the objects are over-segmented (dilation), it tends to have smaller effect compared to under-segmentation. This is likely because over-segmentation still provides full information about an object. More generally, SCAT is still performs better or similar to SiS when the kernel sizes of dilation and erosion is relatively small (9 for dilation and 7 for erosion). The implication is that even the segmentation model makes small errors, explicit models like SCAT will still outperform single slot models.

Although the main focus of our work is on measuring the benefit of object-centric video modeling in a controlled setting, we also compare our method with other similar methods to better contextualize those results. Our model is designed to be small yet efficient, demonstrating high performance without the need for large-scale resources. In contrast, many existing models rely on significantly larger architectures and large-scale datasets to achieve similar results, which can be resource-intensive and less practical. To ensure a fair and balanced evaluation, we therefore adjusted each method's hyperparameters to match our model's size (i.e., number of weights), providing a level playing field for comparison. We compare against VideoGPT (Yan et al., 2021), which uses a similar architecture, and the CNN-based SimVP (Gao et al., 2022) for a comprehensive evaluation. Prediction performance on **KTH**, **Real-Traffic** and **Kubric-Real** are

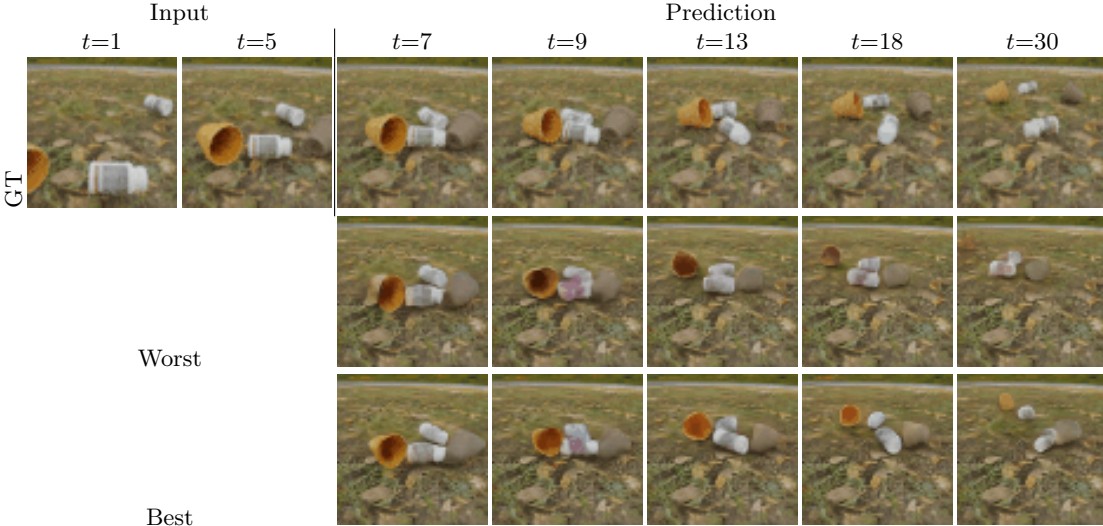

Figure 9: Qualitative results of worst and best cases of 25 samples generated by SCAT on Kubric-Real dataset when the temperature equals to 0.7 (best among all temperatures)

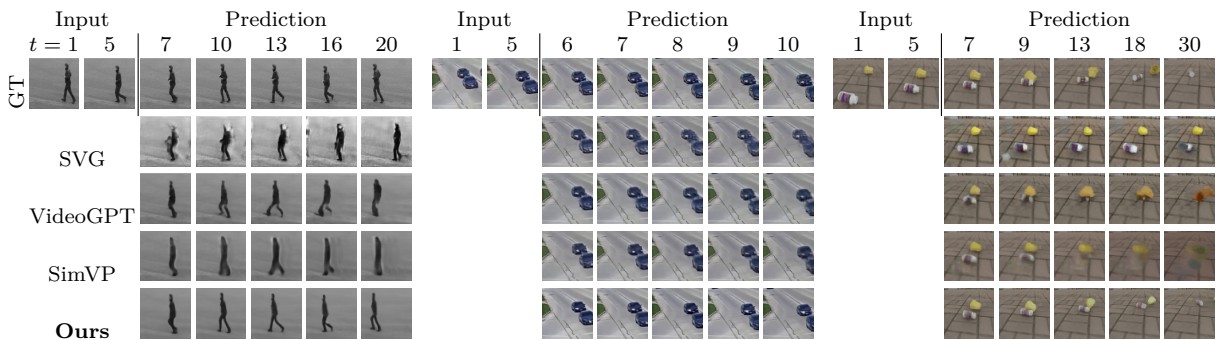

Figure 10: Qualitative results from our full model and baselines on **KTH** (left), **Real-Traffic** (middle) and **Kubric-real** (right).

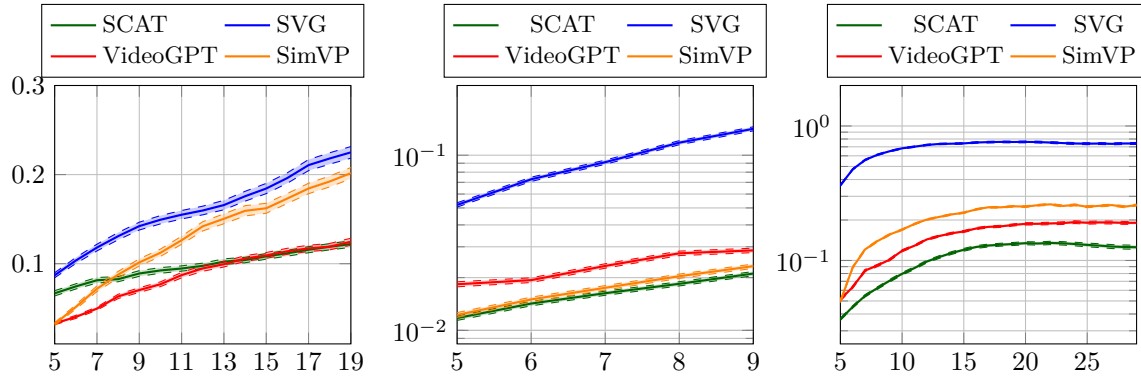

Figure 11: Mean and Std of LPIPS metric for **KTH(left)**, **Real-Traffic(middle)** and **Kubric-Real(right)** datasets, where x-axis and y-axis denotes time-step and mean±std, respectively.

presented in Table 3 and Figure 11. The SCAT model outperforms or is competitive with other models across all three datasets, with a smaller model size, confirming the effectiveness of instance-level segmentation and cross-attention. On the simpler **KTH** dataset, SCAT achieves same SSIM compared to VideoGPT (0.789

Table 3: Quantitative results on **KTH**, **Real-Traffic** and **Kubric-Real** datasets

| | KTH | | | | Real-Traffic | | | | Kubric-Real | | | |
|---|---|---|---|---|---|---|---|---|---|---|---|---|
| | PSNR↑ | SSIM↑ | LPIPS↓ | Num-Prms | PSNR↑ | SSIM↑ | LPIPS↓ | Num-Prms | PSNR↑ | SSIM↑ | LPIPS↓ | Num-Prms |
| SVG | 15.93±0.23 | 0.614±0.008 | 0.161±0.004 | 23M | 25.64±0.11 | 0.900±0.002 | 0.095±0.0024 | 31M | 16.52±0.13 | 0.611±0.006 | 0.699±0.009 | 41M |
| VideoGPT | 24.44±0.18 | 0.789±0.004 | **0.087±0.002** | 41M | 29.13±0.10 | 0.927±0.001 | 0.023±0.0006 | 55M | 23.62±0.17 | 0.700±0.005 | 0.155±0.003 | 67M |
| SimVP | 25.17±0.22 | **0.812±0.005** | 0.130±0.004 | 32M | 30.16±0.11 | 0.949±0.001 | 0.018±0.0004 | 32M | 22.21±0.15 | 0.710±0.005 | 0.213±0.003 | 59M |
| SCAT | **26.54±0.18** | 0.789±0.004 | 0.097±0.003 | 23M | **30.41±0.12** | 0.949±0.001 | **0.016±0.0004** | 28M | **25.13±0.19** | **0.789±0.004** | **0.108±0.003** | 40M |

Table 4: Comparison of FLOPs (GMac) of a single pass, Peak gRAM (GB) and Latency (s) of finish predicting set of frames (15 for KTH, 5 for Real-Traffic, 25 for Kubric-Real)

| Dataset | SVG | | | VideoGPT | | | SimVP | | | SCAT | | |
|---|---|---|---|---|---|---|---|---|---|---|---|---|
| | FLOPs | Peak gRAM | Latency | FLOPs | Peak gRAM | Latency | FLOPs | Peak gRAM | Latency | FLOPs | Peak gRAM | Latency |
| KTH | 28.92 | 0.46 | 0.31 | 34.96 | 1.27 | 15.03 | 10.5 | 0.80 | 0.04 | 73.1 + (46.6) | 0.68 + (0.98) | 0.61 + (0.36) |
| Real-Traffic | 1.22 | 0.50 | 0.14 | 35.10 | 1.46 | 6.29 | 10.5 | 0.57 | 0.01 | 291.1 + (344.1) | 1.05 + (1.19) | 1.33 + (1.94) |
| Kubric-Real | 48.09 | 0.55 | 0.49 | 35.21 | 1.64 | 26.23 | 24.2 | 1.49 | 0.12 | 322.5 + (0.0) | 1.25 + (0.0) | 4.06 + (0.0) |

vs 0.789) and slightly lower LPIPS than VideoGPT (0.087 vs 0.097), but lower quality according to PSNR (26.54 vs 24.44). Moreover, from Figure 10 we can see that only SCAT maintained human posture throughout the prediction. On **Real-Traffic**, SCAT achieved best performance in PSNR metric, with PSNR of 30.41, which is higher than VideoGPT (29.13) and SimVP(30.16). Moreover, SCAT also performs best under the perceptually robust LPIPS metric (0.016), outperforming both VideoGPT (0.023) and SimVP (0.018), indicating better perceptual quality. Also, from Figure 10 we can see that when $t$=9 and $t$=10, SCAT maintained the distance between two cars and kept them separate while the other models merged the two cars. Finally, on **Kubric-Real**, where strong interactions and realistic objects are present, our model leads by a large margin on every metric. This further demonstrates that the proposed model achieves larger improvements on scenes with more instances and strong interactions. In Figure 10, SimVP, VideoGPT and SVG all failed to predict the collision between two objects, while SCAT predicted this accurately and maintained the object shape.

Following internal experiments, we also compare the proposed method in terms of computational efficiency against the baselines. We compare FLOPs of a single forward pass, peak GPU memory usage in inference time and the total time spent to finish predicting the required number of frames for a dataset. From Table 4 we can see that in all datasets, SCAT's FLOPs are the highest among other baselines, and it is scaled up further with the addition of segmentation models. This is an expected limitation of our model that as the number of classes and instances increases, the cross-attention module will be operated between each instance pairs, leading to high computational cost. It is still worth noting that all of the experiments conducted in this paper used relatively limited computation power (single NVIDIA RTX 3090 GPU), therefore this approach can be scaled to devices having more computation power to potentially scale up the inference latency.

## 5 Conclusion

In this paper, we investigated and analyzed the benefits of explicit object-centric decomposition in video prediction. We presented a flexible video prediction pipeline based on an object-aware VQ-VAE and multi-object Transformer, that operates on separate objects extracted via panoptic segmentation; we also defined variants that lack object-decomposition and support for interactions to measure the impact of these design choices in a controlled manner. We evaluated the proposed models on five datasets, finding that when a dynamic scene is explicitly decomposed and encoded into a structured latent vector, prediction quality is better than an equal-capacity model without decomposition, and that this improvement is larger for scenes that involve strong interactions between objects. This confirms that using both object decomposition and cross-attention to handle interactions improves the overall prediction quality when strong interactions occur in a dynamic scene.

# 6 Limitations

Our model has three inherent limitations. First, object decomposition is entirely reliant on the performance of instance segmentation models, this is evident in Figure 8 that the proposed model's performance is decreased when the kernel sizes to simulate over- and under-segmentation became bigger. Second, our experiments throughout the paper focused solely on static camera settings, and additional experiments would be required to evaluate the robustness of the approach to scenarios with moving cameras. Third, the encoder encodes predefined object classes. For example, pots and bottles in Kubric, cars in Real-traffic and spheres in CLEVR datasets. Based on this predefined latent space, the transformer will also learn to predict the dynamics of the given latent space during training. Because each object in a video is first segmented and the instances which belong to the predefined classes are selected to process, if there are novel object classes outside the scope of the predefined classes, then the novel objects are automatically categorized to the background slot. Therefore, this novel object's motion is learned and predicted implicitly. For example in Kubric-Real, the model is trained to predict the motions of pots and bottles, and if we initialize a new object with different characteristics than pre-defined object-class (i.e., a box), its motion is learned in the background slot implicitly.

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

# Appendix

## A    Implementation Details

Specific implementation details of both **Object Aware Auto-Encoder (OAAE)**, **Stochastic Class-Attended Transformer (SCAT)** and their variants are given in Tables 5 and 6. We ensured that the non-decomposed version was fairly compared to the decomposed version by adjusting the embedding dimensions accordingly. Specifically, the embedding dimension in the non-decomposed version was set to be $N$ times larger than the embedding dimension of a single instance in the decomposed setting, where $N$ represents the total number of instances. For example, in the **Kubric-Real** dataset, there are three classes: background, bottles, and pots. The background class is assigned one slot, the bottles class is assigned two slots, and the pots class is assigned two slots, totalling five instances. Thus, if each instance in the decomposed version has an embedding dimension of 128, then in the non-decomposed version, the embedding dimension is set to 640 (128 times 5 instances).

| | KTH | | Real-Traffic | | CLEVR-2 | | CLEVR-3 | | Kubric-Real | |
|---|---|---|---|---|---|---|---|---|---|---|
| | OAAE | Non-Decom | OAAE | Non-Decom | OAAE | Non-Decom | OAAE | Non-Decom | OAAE | Non-Decom |
| In Channels | 1 | | 3 | | 3 | | 3 | | 3 | |
| Num Instance | 2 | 1 | 5 | 1 | 3 | 1 | 4 | 1 | 5 | 1 |
| Num Classes | 2 | 1 | 2 | 1 | 2 | 1 | 2 | 1 | 3 | 1 |
| Embed Dim/Instance | 128 | 256 | 128 | 640 | 128 | 384 | 128 | 512 | 128 | 640 |
| Num Embeddings | 5120 | | 5120 | | 5120 | | 5120 | | 5120 | |
| Conv Hidden Dims | 128, 256 | | 128, 256 | | 128, 256 | | 128, 256 | | 128, 256 | |
| Num Residual Layers | 6 | | 6 | | 6 | | 6 | | 6 | |
| Batch Size | 8 | | 8 | | 8 | | 8 | | 8 | |
| Learning Rate | $10^{-4}$ | | $10^{-4}$ | | $10^{-4}$ | | $10^{-4}$ | | $10^{-4}$ | |

Table 5: Hyper-parameters of OAAE on **KTH**, **Real-Traffic**, **CLEVR-2**, **CLEVR-3** and **Kubric-Real** Datasets

## B    Dataset Details

### B.1    Decomposition

For KTH, we use CLIPSeg with the prompt `'person'` and `'background'` to decompose the frames. For Real-Traffic, we use YOLOv8 to be our instance segmentor. For Kubric-generated datasets, because the instance segmentation map is available with the generation, we directly use these to extract the instances.

### B.2    Synthetic Datasets Generation

We use Kubric to generate **CLEVR-2**, **CLEVR-3** and **Kubric-Real**. The generation parameters are given in Table 7. All three datasets use a colliding position range of $[-1, 1]$ and a fixed, static camera looking at $(0, 0)$. The summoning radius is set to 5 for CLEVR datasets and 8 for Kubric-Real, with minimum summoning distances of 2 for CLEVR and 4 for Kubric-Real. CLEVR datasets feature object friction values of 0.4 for metal spheres and 0.8 for rubber spheres, while Kubric-Real has a uniform friction of 1.0. This higher friction in Kubric-Real necessitates a larger maximum initial velocity of 7, compared to 5 in the CLEVR datasets. The number of objects also increases from 2 in CLEVR-2 to 3 in CLEVR-3, and 4 in Kubric-Real. More details are given in table 7.

## C    Additional Qualitative Results

| | KTH | | | Real-Traffic | | | CLEVR-2 | | | CLEVR-3 | | | Kubric-Real | | |
|---|---|---|---|---|---|---|---|---|---|---|---|---|---|---|---|
| | SCAT | SNCAT | SiS | SCAT | SNCAT | SiS | SCAT | SNCAT | SiS | SCAT | SNCAT | SiS | SCAT | SNCAT | SiS |
| Num Instance | 2 | 1 | | 5 | 1 | | 3 | 1 | | 4 | 1 | | 5 | 1 | |
| Num Classes | 2 | 1 | | 2 | 1 | | 2 | 1 | | 2 | 1 | | 3 | 1 | |
| VQVAE Dim | 128 | 256 | | 128 | 640 | | 128 | 384 | | 128 | 512 | | 128 | 640 | |
| Embed Dim/Instance | 256 | 512 | | 256 | 1280 | | 256 | 768 | | 256 | 1024 | | 256 | 1280 | |
| Num Attention Head | 16 | | | 16 | | | 16 | | | 16 | | | 16 | | |
| FF expanding Factor | 2 | | | 2 | | | 2 | | | 2 | | | 2 | | |
| Depth | 4 | | | 4 | | | 4 | | | 4 | | | 4 | | |
| Drop Out | 0.3 | | | 0.3 | | | 0.3 | | | 0.3 | | | 0.3 | | |
| Batch Size | 1 | | | 1 | | | 1 | | | 1 | | | 1 | | |
| Learning Rate | $10^{-4}$ | | | $10^{-4}$ | | | $10^{-4}$ | | | $10^{-4}$ | | | $10^{-4}$ | | |
| LR Scheduler | Cosine | | | Cosine | | | Cosine | | | Cosine | | | Cosine | | |
| Warm-up Steps | 10000 | | | 10000 | | | 10000 | | | 10000 | | | 10000 | | |

Table 6: HyperParameters of SCAT and its variants on **KTH**, **Real-Traffic**, **CLEVR-2**, **CLEVR-3** and **Kubric-Real** Datasets

| | CLEVR-2 | CLEVR-3 | Kubric-Real |
|---|---|---|---|
| Colliding Position Range $(x, y)$ | [(-1, 1),(-1, 1)] | [(-1, 1),(-1, 1)] | [(-1, 1),(-1, 1)] |
| Radius for Summoning Objects | 5 | 5 | 8 |
| Min Distance When Summoning | 2 | 2 | 4 |
| Max Initial Velocity | 5 | 5 | 7 |
| Ground Friction | 0.3 | 0.3 | 0.3 |
| Object Friction | 0.4,0.8 | 0.4,0.8 | 1.0 |
| Num Objects | 2 | 3 | 4 |
| Num Object Class | 1 | 1 | 2 |
| Camera Position | Fixed Static | Fixed Static | Fixed Static |
| Camera Looks At $(x, y, z)$ | (0, 0, 0) | (0, 0, 0) | (0, 0, 0) |

Table 7: Parameters for Generating **CLEVR-2**, **CLEVR-3** and **Kubric-Real** Datasets

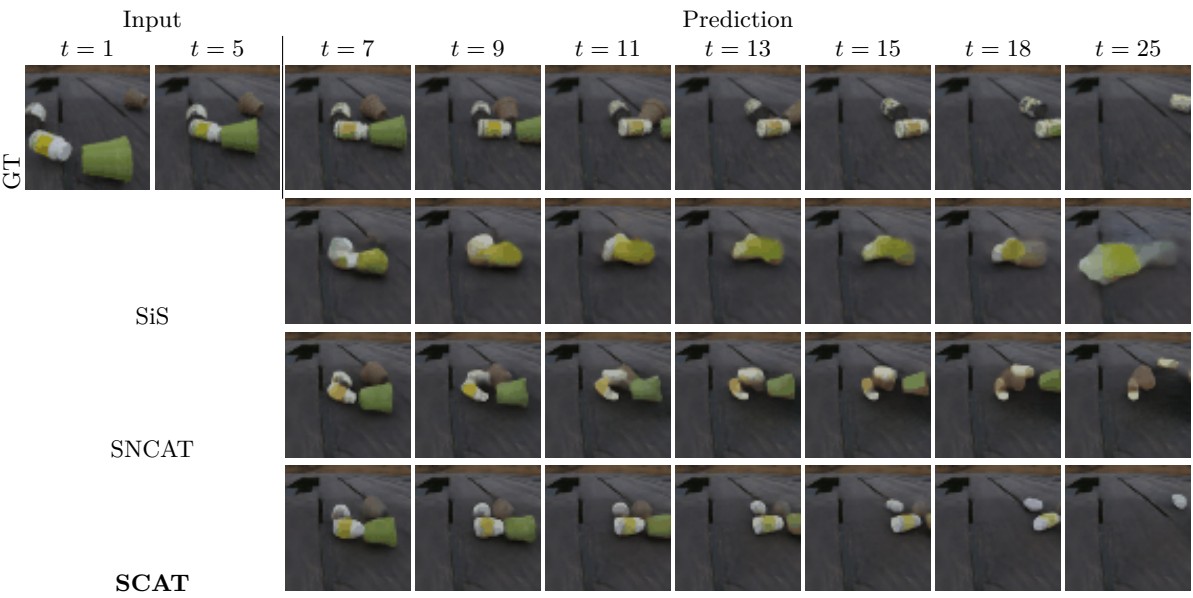

Figure 12: **Kubric-Real** Example 1

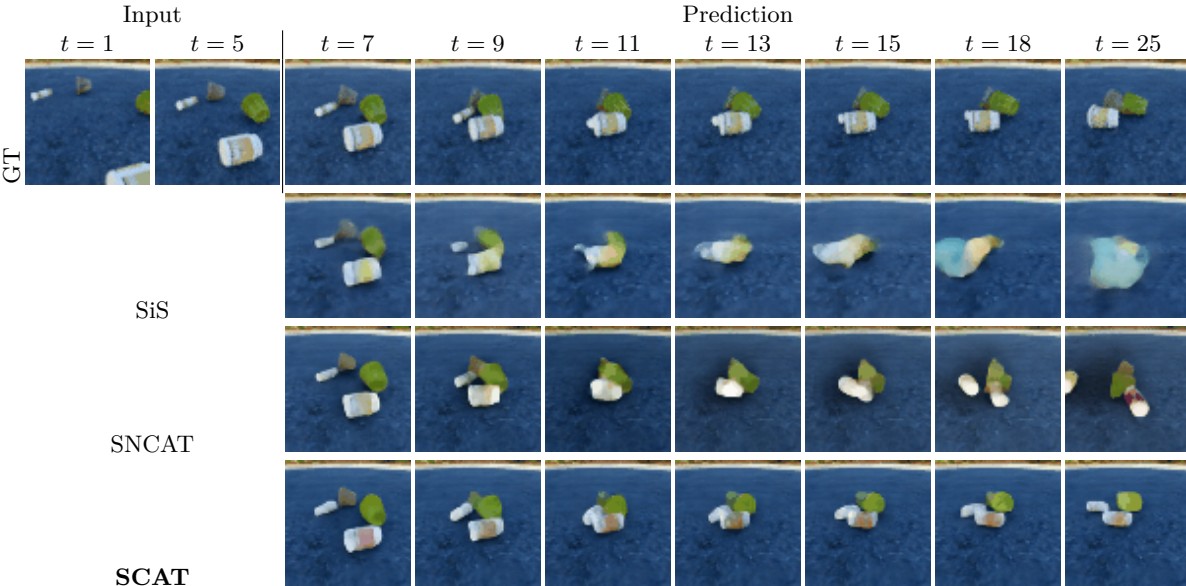

Figure 13: **Kubric-Real** Example 2

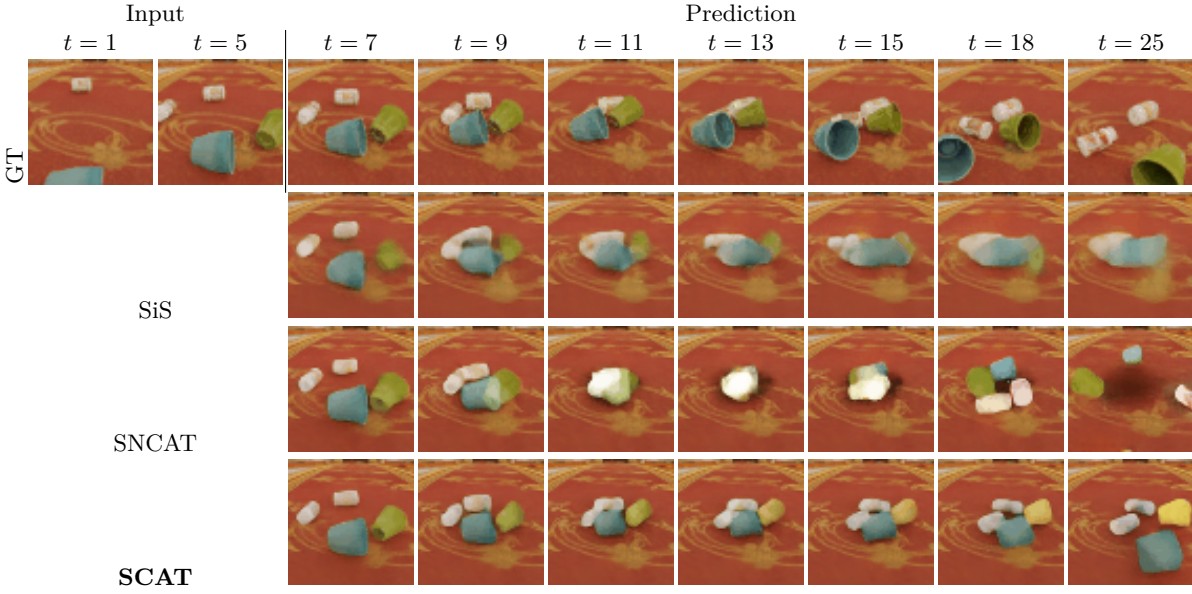

Figure 14: **Kubric-Real** Example 3

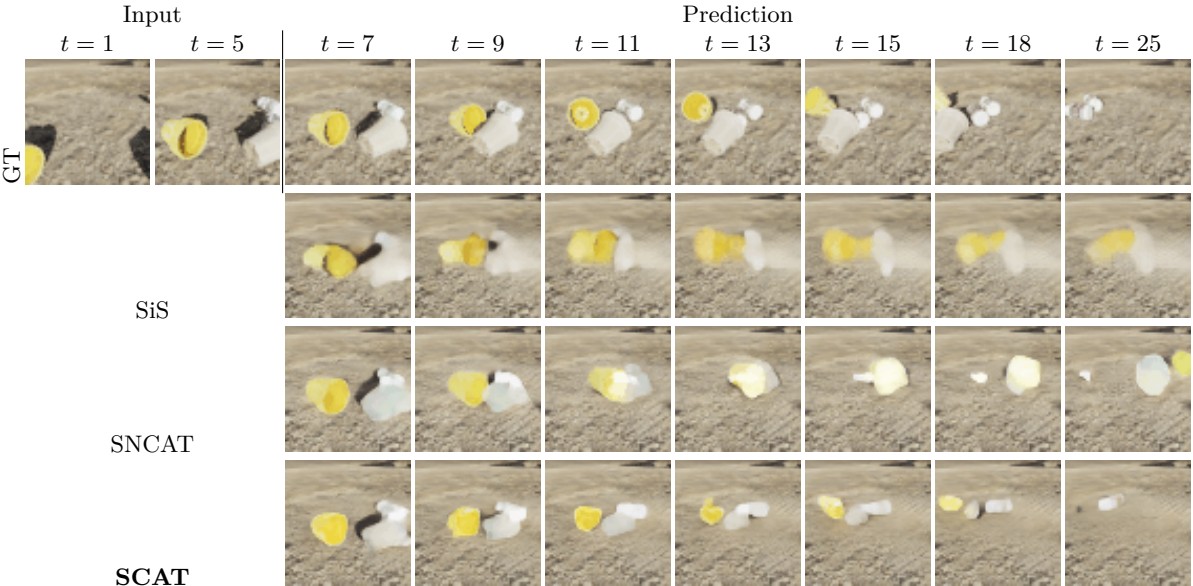

Figure 15: **Kubric-Real** Example 4

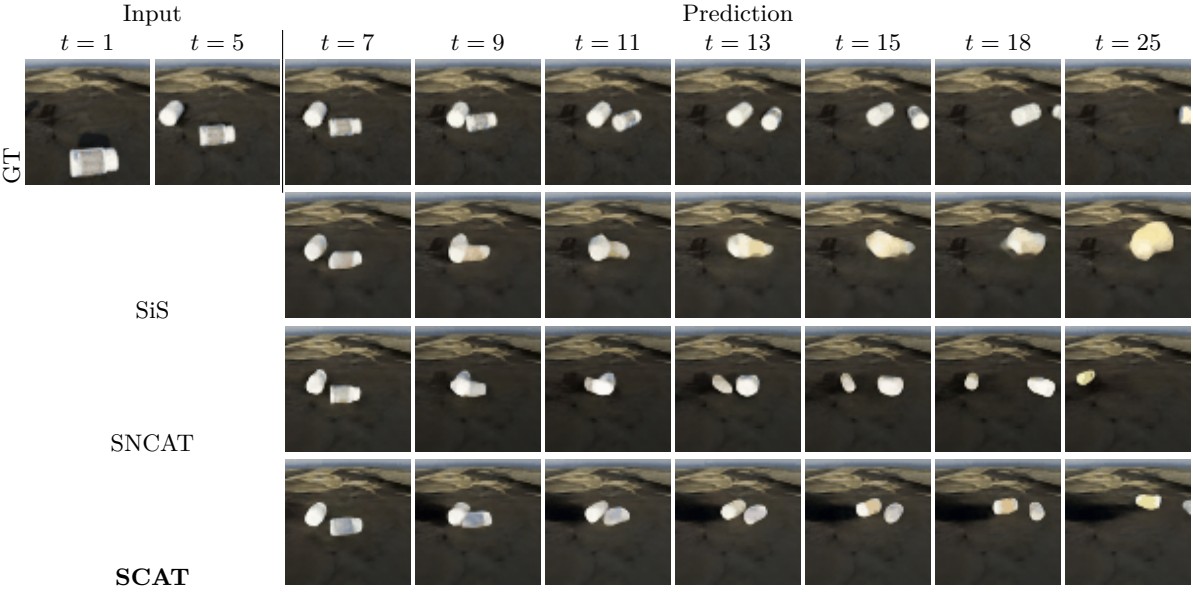

Figure 16: **Kubric-Real** Example 5

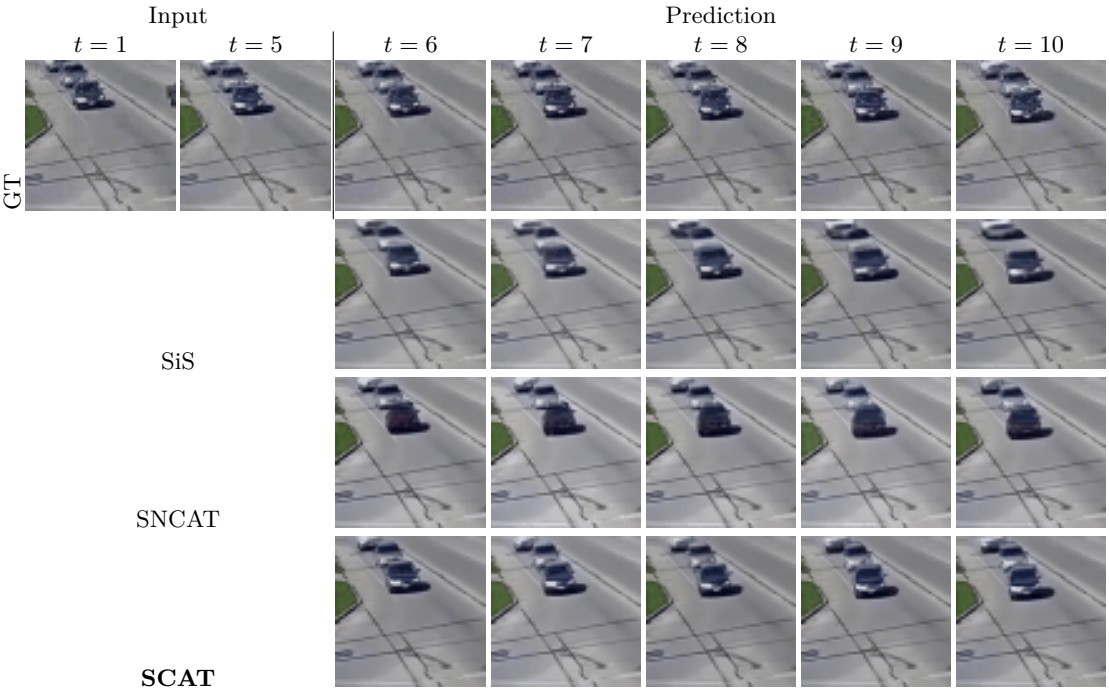

Figure 17: **Real-Traffic** Example 1

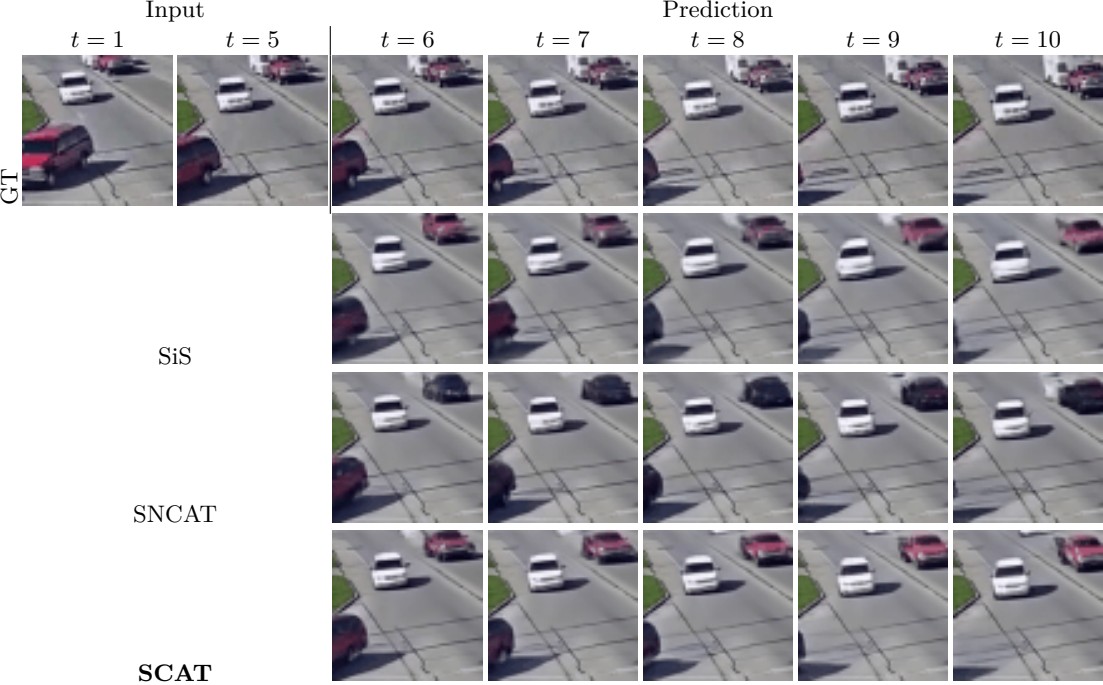

Figure 18: **Real-Traffic** Example 2

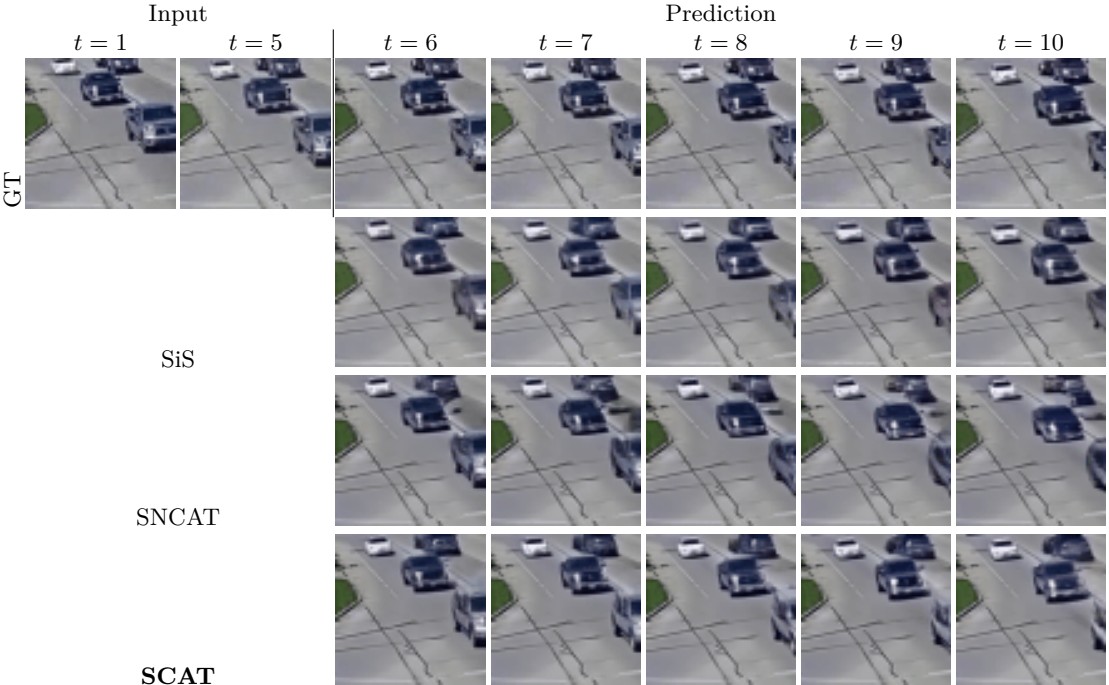

Figure 19: **Real-Traffic** Example 3

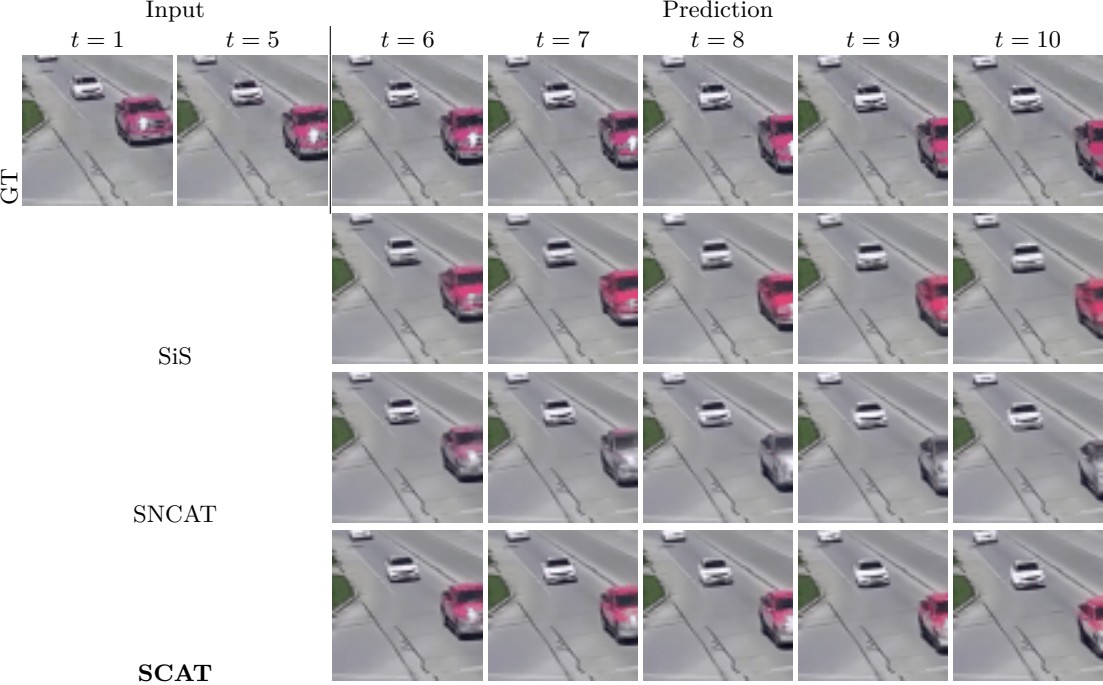

Figure 20: **Real-Traffic** Example 4

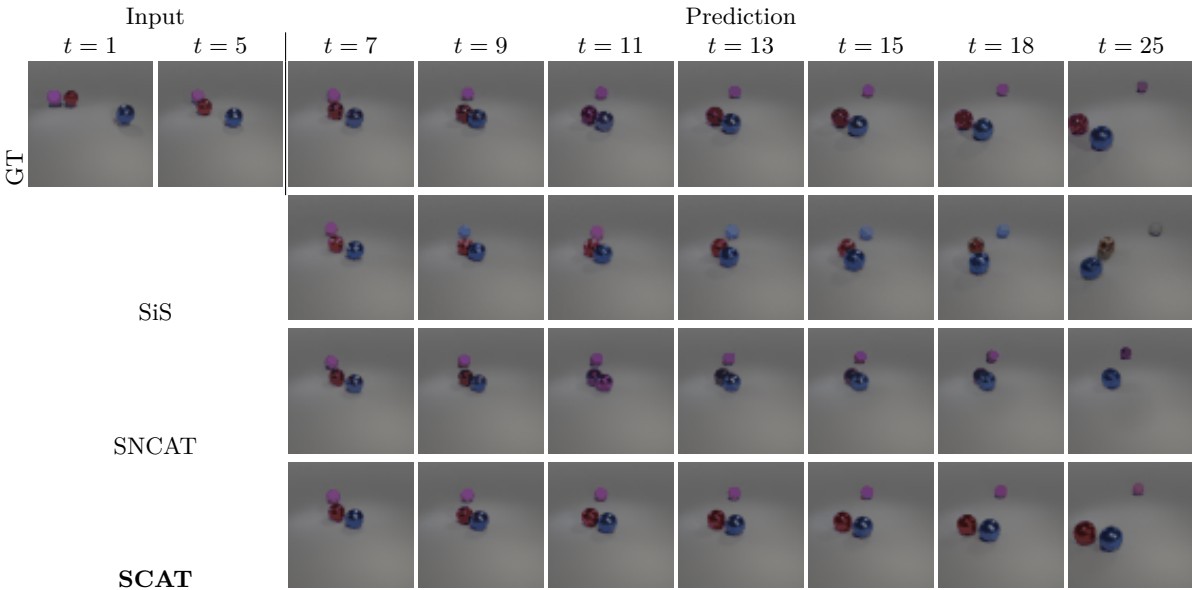

Figure 21: **CLEVR-3** Example 1

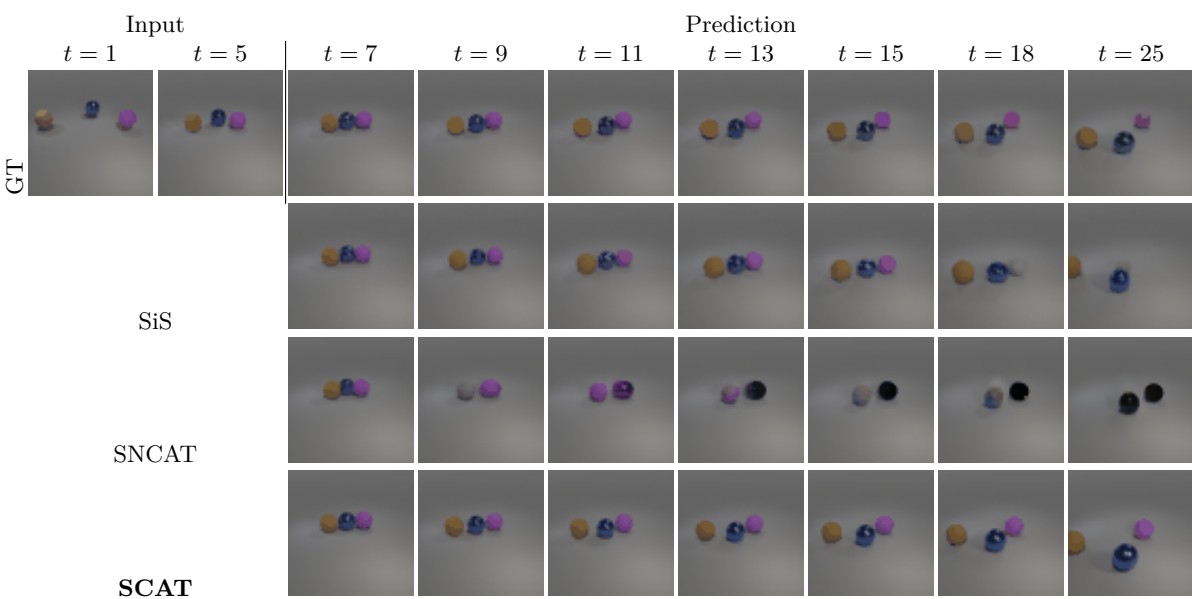

Figure 22: **CLEVR-3** Example 2

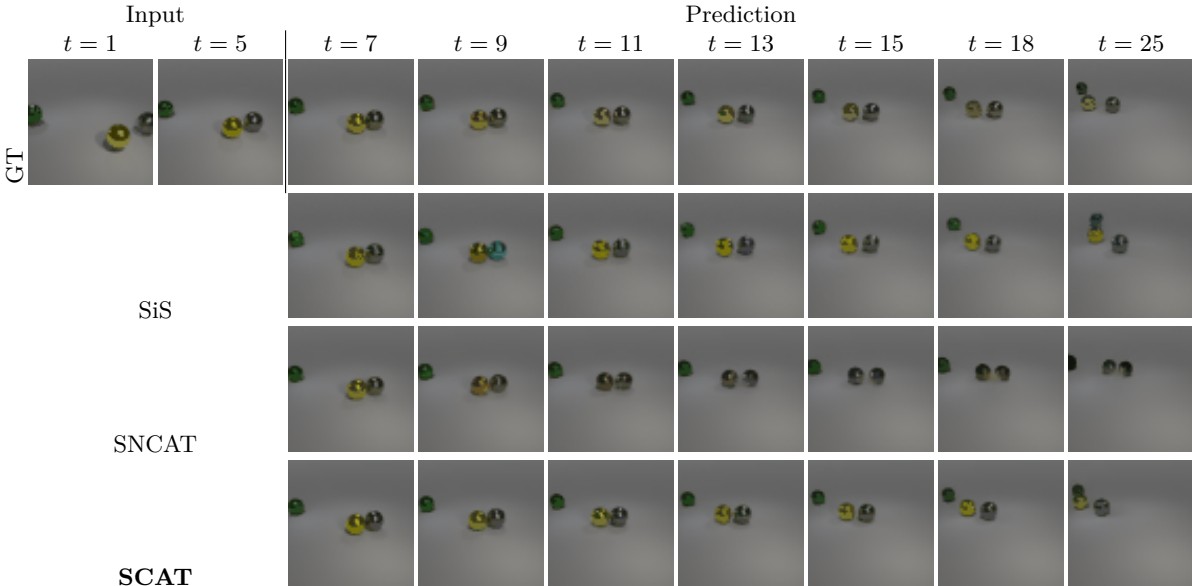

Figure 23: **CLEVR-3** Example 3

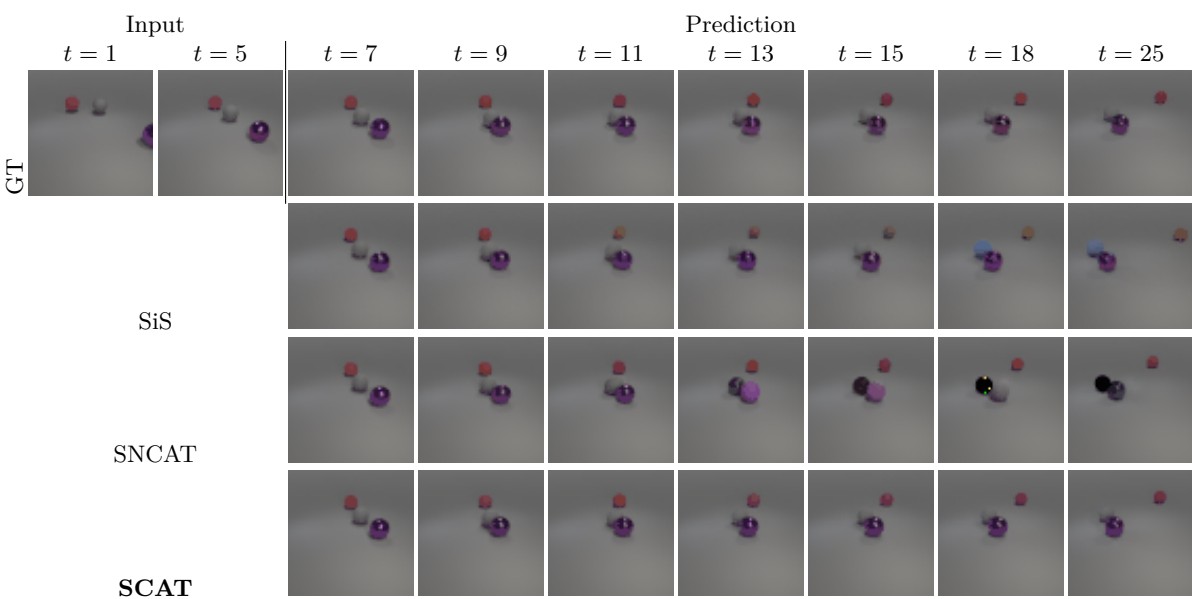

Figure 24: **CLEVR-3** Example 4

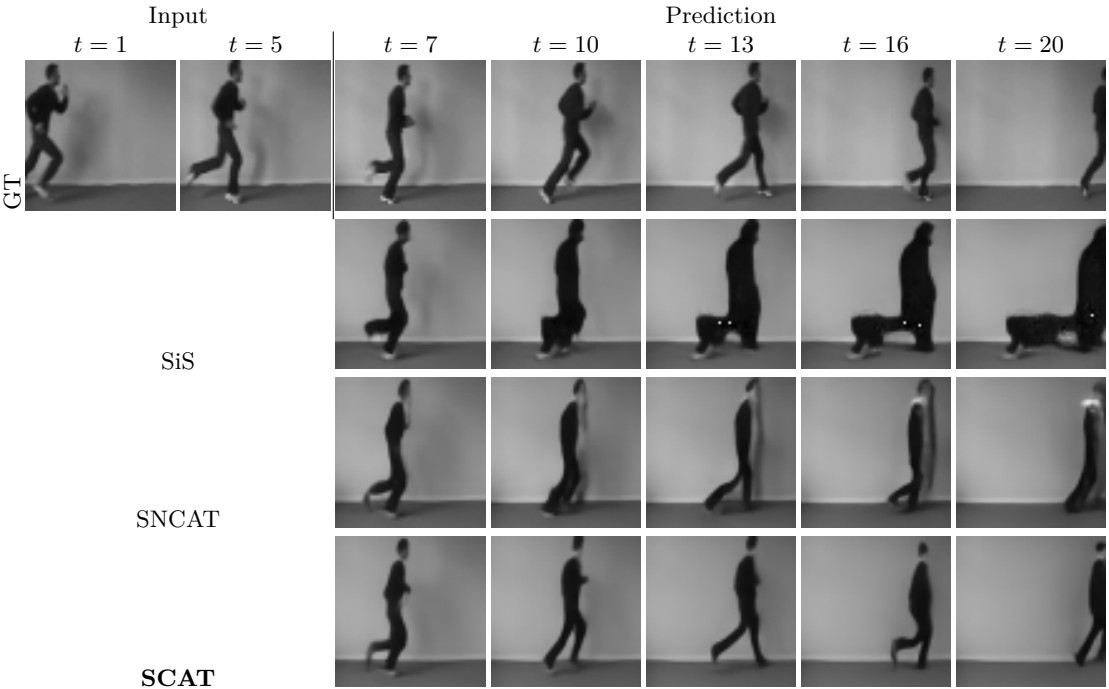

Figure 25: **KTH** Example 1

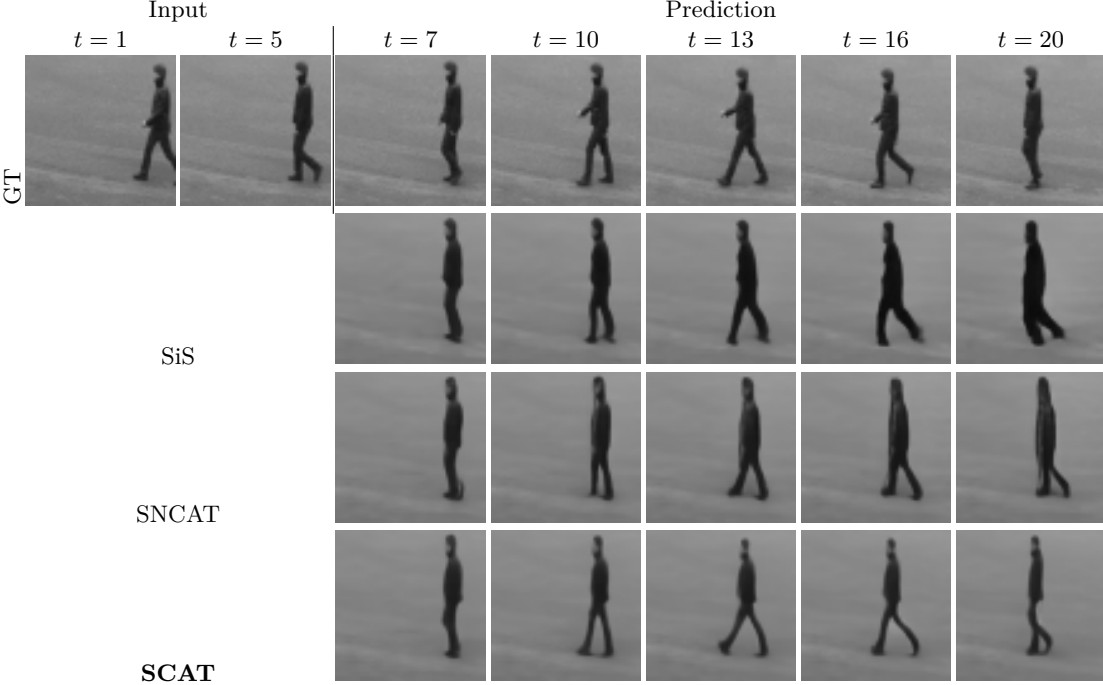

Figure 26: **KTH** Example 2

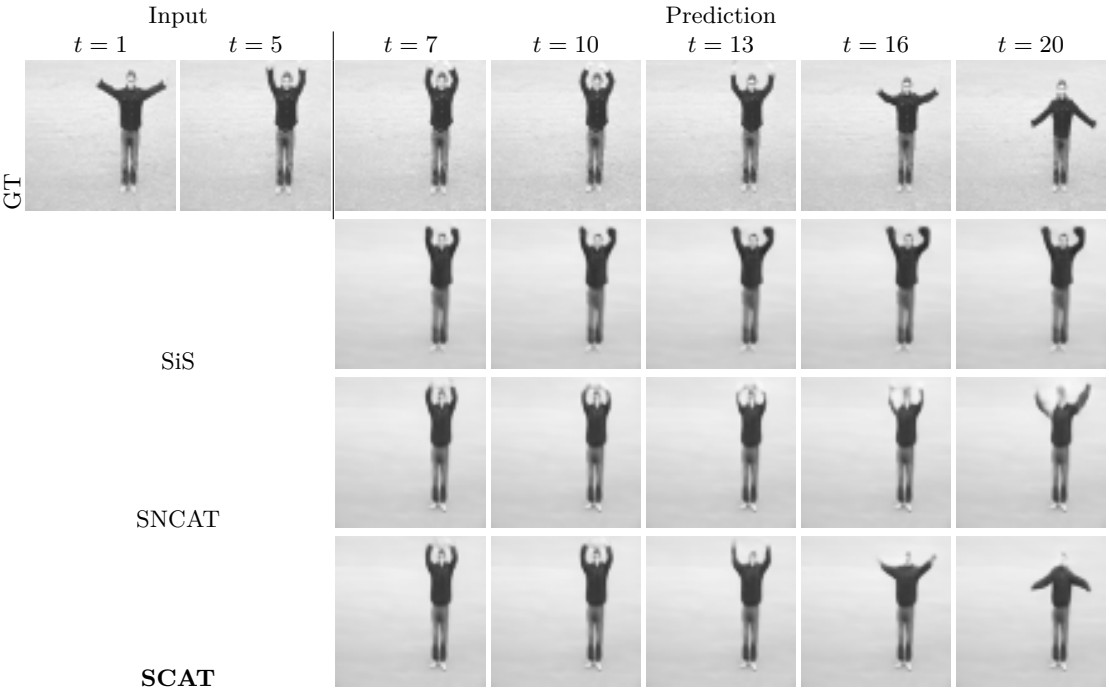

Figure 27: **KTH** Example 3

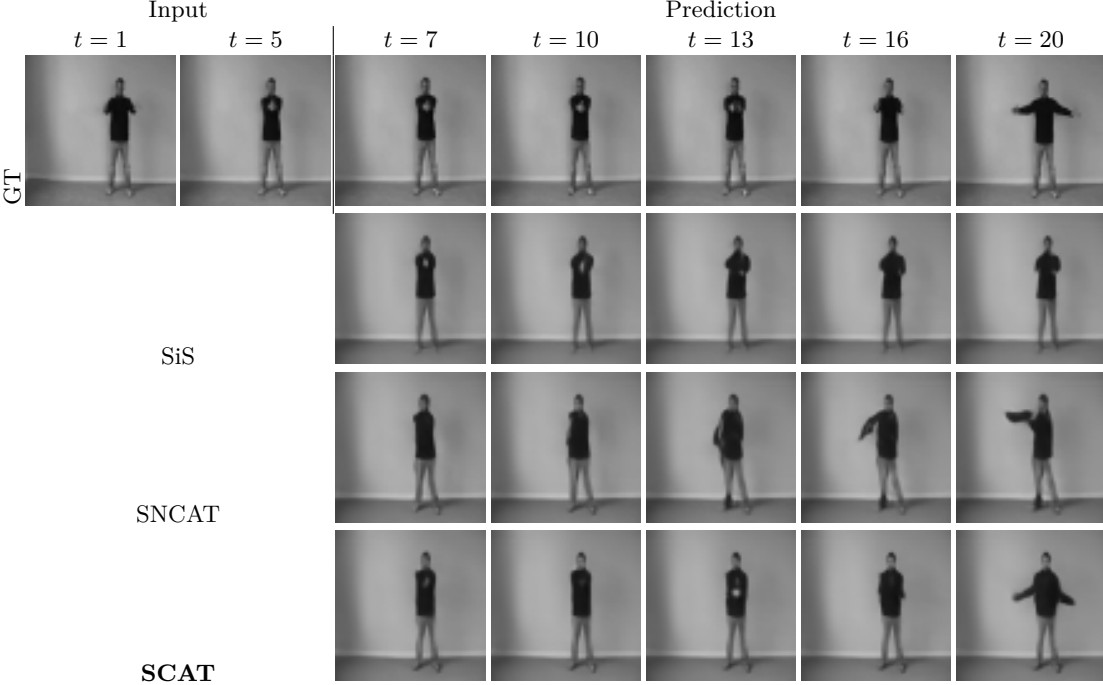

Figure 28: **KTH** Example 4

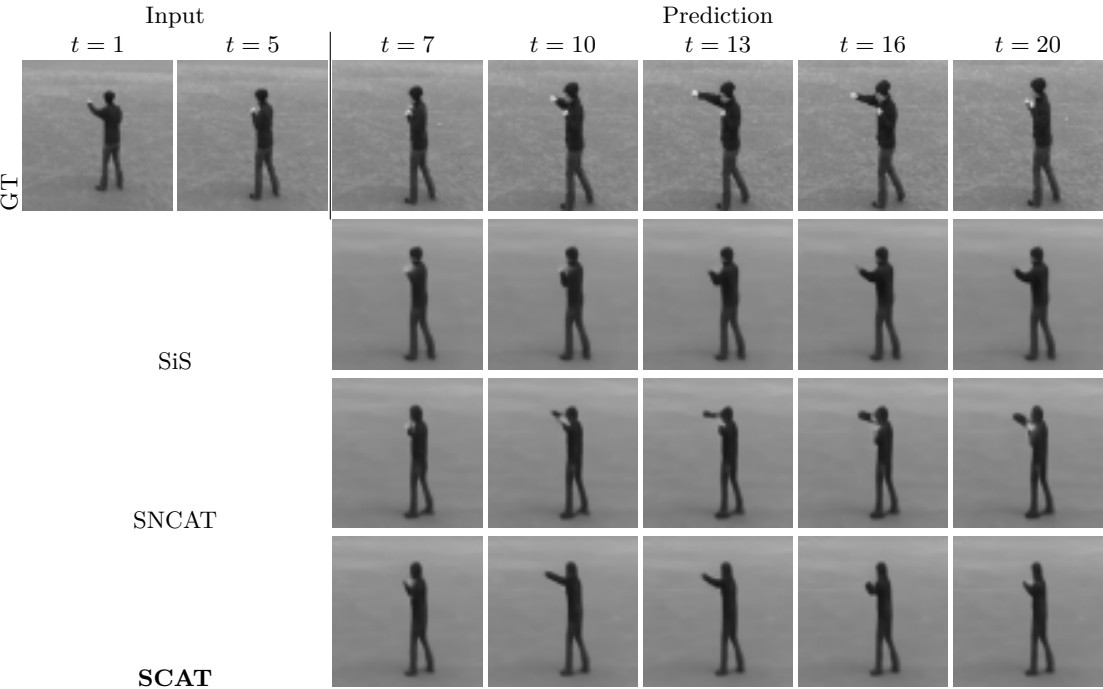

Figure 29: **KTH** Example 5

