# OpenReview forum: "On the Benefits of Instance Decomposition in Video Prediction Models"
_TMLR — Rejected by TMLR_

### Review · Reviewer_F9zv · 2025-08-01

**Summary Of Contributions:**

This paper provides a compelling demonstration of why decomposing scenes into individual objects is a superior strategy for video prediction. The authors use a unified framework, combining a VQ-VAE-based encoder with a multi-slot Transformer, to systematically test this idea. Through a series of well-chosen experiments, they show that their object-centric approach, especially when modeling interactions, yields significantly better and more physically plausible predictions.

### Strengths
- Well-controlled experimental design: the authors frame their work around a clear, testable hypothesis. The methodical comparison between the SiS, SNCAT, and SCAT model variants provides an interpretable ablation study that successfully isolates the benefits of instance decomposition and interaction modeling. The results are compelling and strongly support the central thesis that an object-centric approach is highly beneficial for video prediction.
- The writing is well-structured and insightful, easy to follow.

### Limitations
While the core contribution is strong and the paper is well-written, I believe its value could be significantly enhanced by explicitly addressing its key limitations. My primary concerns are related to the framework's applicability.

- The framework's success is predicated on the scene being cleanly decomposable into discrete instances. This is a key assumption that should be stated as a limitation, as it restricts the method's use in domains without clear "objects" (e.g., fluid dynamics, weather patterns) and makes its performance dependent on the quality of an upstream segmentation model.
- All experiments are conducted with a static camera, which simplifies the problem by removing the challenge of camera ego-motion. This is a significant limitation that currently prevents the method's direct application to key domains like autonomous driving (e.g., on KITTI/Cityscapes). Acknowledging this would help frame the work as a foundational step that future research can build upon to tackle more complex, real-world scenarios.

**Audience:**

Yes

**Audience Explanation:**

The paper offers a strong, convincing argument that object-centric approaches are not just a niche idea but a superior strategy, especially for scenes with complex dynamics. Researchers designing future video prediction models will likely use this insight to guide their own architectural choices.

**Claims And Evidence:**

Yes

**Claims Explanation:**

The evidence directly supports their two main claims in a controlled way.
- The claim that decomposition itself is beneficial is proven by the direct comparison between their holistic model (SiS) and the decomposition-only model (SNCAT). This cleanly isolates the variable and shows its effect.
- Similarly, the claim that modeling interactions provides a further, critical boost is backed by the head-to-head comparison between SNCAT (no interaction) and SCAT (with interaction).

**Requested Changes:**

Add a "Limitations" Section:
- Dependency on Upstream Segmentation: The authors should explicitly state that the framework's performance is fundamentally reliant on the quality of an external instance segmentation model. This includes discussing how the method's applicability is restricted to domains where clear instance decomposition is feasible and acknowledging that segmentation errors would directly impact prediction quality.
- Static Camera Assumption: The paper should clearly state that all experiments were conducted with a static camera and that the current framework does not address the challenge of ego-motion.

Could the author please elaborate on the following points:
Handling of Novel Classes (Open-World Setting): What is the expected behavior of the model when it encounters an object from a class that was not seen during training? As the framework seems to require a pre-defined encoder for each class, does it have any mechanism to handle novel objects, or would it result in a failure case?

Minor polish: A final proofreading pass is recommended to address formatting issues: the citation placeholder (?) on p. 8.

---

> ### Author Response · Authors · 2025-09-09
> **Response to F9zv**
>
> We thank the reviewer for their thoughtful feedbacks, and we are pleased to know the reviewer finds our experiments are well designed and paper is well written. We address the requests and concerns of the reviewer below.
>
> - "***Add limitation section***":
>
>      A limitation section is added, that states explicitly the limitations of the proposed methods:
>     > "   Our model has three inherent limitations. First, object decomposition is entirely reliant on the performance of instance segmentation models, this is evident in Figure 8 that the proposed model's performance is decreased when the kernel sizes to simulate over- and under-segmentation became bigger. Second, our experiments throughout the paper focused solely on static camera settings, and additional experiments would be required to evaluate the robustness of the approach to scenarios with moving cameras. Third, the encoder encodes predefined object classes. For example, pots and bottles in Kubric, cars in Real-traffic and spheres in CLEVR datasets. Based on this predefined latent space, the transformer will also learn to predict the dynamics of the given latent space during training. Because each object in a video is first segmented and the instances which belong to the predefined classes are selected to process, if there are novel object classes outside the scope of the predefined classes, then the novel objects are automatically categorized to the background slot. Therefore, this novel object's motion is learned and predicted implicitly. For example in Kubric-Real, the model is trained to predict the motions of pots and bottles, and if we initialize a new object with different characteristics than pre-defined object-class (i.e., a box), its motion is learned in the background slot implicitly. "
>
> - "***What is the expected behaviour of the model when it encounters an object from a class that was not seen during training?***":
>
>     Since our model uses pre-defined object-classes and slots, therefore when a novel object appears in the scene, it will be categorized as part of the background and all of the objects in the background slot will perform similarly as SiS model variant which the motion is learned implicitly.
>
> - All of the minor typos and formatting issues are addressed as requested.

---

### Review · Reviewer_CArN · 2025-08-17

**Summary Of Contributions:**

### Summary
The paper studies object-centric video prediction by decomposing each frame into per-object latents and forecasting them with a transformer. An object-aware autoencoder (OAAE) produces a latent for each detected instance (via a pretrained segmenter). The authors compare: SiS (single latent “slot,” no decomposition), SNCAT (per-instance slots with class-shared parameters, no cross-instance attention), and SCAT (adds cross-instance attention). Experiments on KTH, Real-Traffic, and Kubric-based synthetic datasets evaluate PSNR/SSIM/LPIPS under size-matched settings. The main findings are that (i) decomposition plus cross-instance attention improves prediction when objects interact, and (ii) SCAT achieves competitive quality at lower parameter counts than baselines.

### Strengths
- Parameter efficiency. SCAT is competitive/better than strong baselines with notably fewer parameters (e.g., on Real-Traffic).
- Clear ablations. The SiS → SNCAT → SCAT ladder isolates the effects of decomposition and cross-instance attention; the OAAE design is straightforward and reproducible.

### Weaknesses
- Assumed class/slot knowledge (critical). The method assumes known object classes and a fixed slot budget per video. In real scenes, classes may be unknown and object counts vary over time; without tracking, this risks slot overflow, idle slots, and identity switches.
- End-to-end efficiency unclear (critical). SCAT/SNCAT require per-frame segmentation whose runtime/memory can dominate inference, so the “parameter-efficient” claim may overstate practical efficiency without full-pipeline profiling.
- Evaluation scope. Most datasets are constrained or synthetic (Kubric variants, KTH). The paper lacks robustness studies (e.g., variation in synthetic scene parameters) to support broader generalization.
- Reporting choices
  - Metrics (LPIPS/PSNR/SSIM) are named but not defined in the text
  - Results use best-of-25 samples only

**Audience:**

Yes

**Audience Explanation:**

Yes. Object-centric forecasting is active and the class-shared, per-instance formulation is a useful, lightweight pattern others can build on.

**Broader Impact Concerns:**

No major ethical risks in scope. If positioning toward traffic/surveillance use, a short note on responsible deployment (data governance, avoiding identification/tracking misuse) would be appropriate.

**Claims And Evidence:**

Yes

**Claims Explanation:**

Yes, with caveats. The ablations and size-matched comparisons support the core claims about decomposition and cross-instance attention. However, the absence of end-to-end cost accounting and robustness analyses (segmentation quality, class/slot misspecification) weakens the case for practical efficiency and real-world applicability.

**Requested Changes:**

### Key Requests
- Make class/slot assumptions explicit and stress-test them. State the assumed number of classes and instance slots per dataset and report robustness when these are misspecified (too few/too many objects), including a simple with-tracking vs. no-tracking comparison and a class-agnostic slotting variant.

- End-to-end efficiency accounting. Report throughput/latency, FLOPs, and peak memory for the full pipeline (segmentation → OAAE encode/decode → transformer) and compare fairly against single-slot and external baselines that do not require segmentation—i.e., explicitly charge SCAT/SNCAT for segmentation cost.

-  Clarify training provenance. State clearly that the predictor is trained from scratch and the segmenters are pretrained.

- Define metrics and balance reporting. Briefly define PSNR/SSIM/LPIPS in the main text (and which direction is better). In addition to best-of-25, report mean-of-25 (or expected score at a fixed temperature).

### Strongly Suggested to Improve Paper
-  Evaluate Segmentation robustness. Quantify sensitivity to mask quality (e.g., simple erosions/dilations, dropouts, label noise).
- Evaluate Synthetic variability. For Kubric, vary scene parameters (object count, textures, lighting) to show stability of gains.

### Minor / Typos
- Replace the placeholder “Kubric (?)” with the proper citation.

---

> ### Author Response · Authors · 2025-09-09
> **Response to reviewer CArN**
>
> We thank the reviewer for their insightful and constructive feedback. Below, we address the comments in detail.
>
> - “***State the assumed number of classes and instance slots per dataset***":
>
>     For every dataset and model variant, class and slot (instance) assumptions have been added to the appendix (see Table 7).
>
> - “***report robustness when these are misspecified (too few/too many objects), including a simple with-tracking vs. no-tracking comparison and a class-agnostic slotting variant***":
>
>     Since the SCAT and SNCAT models are defined and trained with fixed classes and instance slots, they can only handle the pre-specified set of objects. Consequently, if a novel object appears in the scene beyond the number of available slots, it is assigned to the background slot, whose behaviour aligns with that of our SiS model variant. Cases where the scene contains fewer objects than the number of slots occur, for instance, in Real-Traffic and Kubric-Real, where the model is configured with 5 slots and many videos feature fewer than 5 objects, with no impact on performance.
>
> - ***Do end-to-end efficiency accounting***":
>
>     As requested, we analyzed the FLOPs of a single forward pass, peak GPU memory usage and total latency of full prediction of required frames based on different datasets. We added the additional results in the revised paper---see Figure 7 and Table 4.
>
> - "***Clarify the training provenance***":
>
>     We added the requested changes, stating explicitly that both encoder and transformers are trained from scratch in the experimental protocol section, see page 8.
>
> - "***Define metrics and balance reporting***":
>
>     We briefly defined the metrics we used in this paper as requested; see the updated experimental protocol section (page 8).
>
> - "***In addition to best-of-25, report mean-of-25 (or expected score at a fixed temperature).***":
>
>     As requested, best, worst and average (including standard deviation) cases are analyzed for a single sequence, and we showcased the benefit we gain from this sampling strategy in Figures 4 and 9.
>
> - "***Evaluate Segmentation robustness. Quantify sensitivity to mask quality (e.g., simple erosions/dilations, dropouts, label noise).***":
>
>     We simulated the over- and under-segmentation of a segmentation model with dilation and erosion and report the results in Figure 8 of the revised paper.
>
> - "***Evaluate Synthetic variability. For Kubric, vary scene parameters (object count, textures, lighting) to show stability of gains.***":
>
>     The datasets used in this paper differ in their physical parameters, such as friction and initial velocity, across synthetic scenes. The two real-world datasets also exhibit distinct physical characteristics of objects. The model’s performance is observed across these varying motion conditions.
>
> - All minor typos and changes are addressed as requested.

---

### Review · Reviewer_QEQw · 2025-08-17

**Summary Of Contributions:**

The paper proposes a video prediction model that leverages explicit instance/object decomposition of query images to enhance prediction in transformer-based latent models, offering improvements over single-slot models without object decomposition. Using off-the-shelf semantic segmentation for instance decomposition, the authors adopt a variant of the VQ-VAE architecture, incorporating standard losses along with additional ones from prior works, such as focal frequency loss. Video prediction is performed similarly to VQ-VAE’s decoder with certain modifications. Experiments are conducted on the KTH, REAL-traffic, CLEVR-2/3, and Kubric-real datasets, evaluated with LPIPS and PSNR metrics, and compared against VideoGPT and SimVP.

Pros)
- Explicitly using object-centric inputs (via using off-the-shelf segmentation models, though) for transformer input would be reasonable.

Cons)
- The novelty of this work is limited. Many prior studies (e.g., MOSO [2]) have already explored decomposed or slot-based video modeling (albeit without using off-the-shelf segmentation). This paper largely reassembles existing components (segmentation + VQ-VAE + cross-attention) rather than proposing a fundamentally new approach. Moreover, the adopted components themselves are existing, and their integration does not seemt to be tested ablatively to demonstrate whether they work well in combination.

- The paper lacks a clear intuition or justification for why explicitly segmented objects should improve performance. The reviewer believes that stronger experimental evidence is needed to support this claim. This reviewer believes that the Slot Attention [0] could do this implicitly, and with some advanced head (i.e., video prediction head), it might address some disadvantages raised by the authors. Could the authors clarify or argue against this point?

- Unconvinced about the reliance on the semantic segmentation for object decomposition. If the input images are over-segmented or under-segmented, can the model still process them effectively? This reviewer notes that even with prior knowledge of the number of objects or classes, the method may fail under over- or under-segmentation because the segmentation performance depends heavily on the generalizability of the segmentation model. Wouldn’t a single-slot model, which performs this implicitly without such concerns, be a safer alternative?

- No computational cost comparison is provided against single-slot methods that do not rely on explicit segmentation models. Using the segmented images as the encoder input, which would obviously incur significantly higher costs than the original images, could introduce additional expenses (e.g., memory or FLOPS) and make an unfair comparison of the proposed method with competing methods challenging.

- The experimental setup appears limited. Most datasets are synthetic, and some of the experiments seem scaled down in scope from this reviewer's perspective. Why did the authors use downscaled images (e.g., 64×64) for the experiments? The choice of employing CLEVR-2 and CLEVR-3 datasets is also unclear; why not use larger or more diverse datasets with a greater variety of images?

- The experimental evaluation is limited, with only a few comparisons to prior work. For example, SlotFormer [1], MOSO [2] is mentioned but not compared. Furthermore, there are no evaluations against more recent state-of-the-art models. This reviewer notes that several recent methods [3, 4, 5] could be included for comparison.


[0] Object-Centric Learning with Slot Attention, NeuRIPS 2020

[1] SlotFormer: Unsupervised Visual Dynamics Simulation with Object-Centric Models, ICLR 2023

[2] Moso: Decomposing motion, scene and object for video prediction, CVPR 2023

[3] Invariant Slot Attention: Object Discovery with Slot-Centric Reference Frames, ICML 2023

[4] Continuous Video Process: Modeling Videos as Continuous Multi-Dimensional Processes for Video Prediction, CVPR 2024

[5] LARP: Tokenizing Videos with a Learned Autoregressive Generative Prior, ICLR 2025

**Additional Comments:**

Minor typos
- 'i.e.' and 'i.e.,' are used inconsistently."
- Kubric (?) on page 8

**Audience:**

Yes

**Audience Explanation:**

The idea may be of interest, but without sufficient justification, its soundness remains questionable.

**Claims And Evidence:**

No

**Claims Explanation:**

This reviewer believes that the paper does not provide a clear intuition for why explicitly feeding segmented objects into Transformers should outperform models without segmentation, leaving the authors’ claim insufficiently supported. Furthermore, relying solely on precision-based metrics without reporting computational cost and restricting evaluations to small datasets makes the justification for the proposed approach unconvincing.

**Requested Changes:**

1. Please see the above cons for major changes.

2. The difference between SCAT and SNCAT is unclear, as Figure 1 only shows the SCAT architecture. A separate figure distinguishing the two is recommended.

3. The overall figure should be refined, and the experimental setups should be expanded.

---

> ### Author Response · Authors · 2025-09-09
> **Response to reviewer QEQw**
>
> We thank the reviewer for their review and insightful suggestions. Here we address the comments of the reviewer as detailed below.
>
> - "***The novelty of this work is limited***":
>
>     The novelty of this work is in the systematic study of the impact of instance decomposition on video prediction. Although previous models cited in the article did use decomposition approaches for video prediction, this is the first systematic analysis of decomposition-based video prediction models. We clarified this in the introduction. Please see the highlighted text in page 2.
>
> - "***integration does not seem to be tested ablatively to demonstrate whether they work well in combination***":
>
>     This ablation is investigated by comparing the SiS (no decomposition), SNCAT (decomposition but no interaction) and SCAT (decomposition and interaction) variants of the same architecture in a controlled and systematic way. We have clarified this point in the methodology, please refer to page 4.
>
> - "***The paper lacks a clear intuition or justification for why explicitly segmented objects should improve performance***":
>
>     The intuition and motivation of this paper is revisited and has been made clearer in the introduction section as follows:
>     > "In this article we experimentally investigate the hypothesis that modelling explicitly the motion of the main objects in a scene and their interaction allows for better video prediction without need for larger models or additional training data."
>
> - "***The SlotAttention could do this implicitly***":
>
>     The primary focus of SlotAttention and our paper is fundamentally different:
>     SlotAttention performs object discovery in an implicit way that iteratively categories a part of features of an image to a slot (object), whereas we study the impact of explicit object decomposition on video prediction.
>
> - "***Unconvinced about the reliance on the semantic segmentation for object decomposition***":
>
>     We use well-established off-the-shelf segmentation models that are widely used and benchmarked in the literatures. Following the reviewer's advice, we added an experiment simulating the importance of accurate segmentation by simulating over- and under-segmentation with dilation and erosion operations. Our results show that our model (SCAT) still performs better than models without decomposition (SiS) even when the segmentation models make notable errors---see Figure 8 in the revised paper.
>
> - "***No computational cost comparison is provided against single-slot methods that do not rely on explicit segmentation models***":
>
>   The computational costs such as FLOPs, Peak GPU memory usage during inference and latency comparisons are added in the result section. The details are updated on the paper in Figure 7 and table 4.
>
> - "***Why did the authors use downscaled images (e.g., 64×64) for the experiments? The choice of employing CLEVR-2 and CLEVR-3 datasets is also unclear; why not use larger or more diverse datasets with a greater variety of images?***":
>
>   We use 64$\\times$64 images to better control the experimental process in a limited computational environment where all models are trained on a single RTX 3090 GPU and the method can be scaled to higher resolutions and larger models with more resources. We use CLEVR-2 and CLEVR-3 for controlled evaluation of decomposition and interaction. CLEVR-2 features collisions between two spheres to test cross-attention in modeling interactions. CLEVR-3 adds a non-interacting sphere to test whether the model learns selective interactions rather than memorizing collision events (see Table 2, Figure 6, appendix).
>
> - "***The experimental evaluation is limited, with only a few comparisons to prior work. For example, SlotFormer [1], MOSO [2] is mentioned but not compared. Furthermore, there are no evaluations against more recent state-of-the-art models***":
>
>   As an analytical study, we selected the most representative and simpler examples of different architectures, for example, we chose SVG for RNN, SimVP for CNN and VideoGPT for latent transformer. We focus mainly on VideoGPT since our method is also based on a latent transformer, and therefore the results are most comparable. Also, our aim is not to propose a new model or outperform state-of-the-art, but to analyze the benefits of explicit object decomposition in a controlled manner. Hence, we do not compare against recent, more complex methods requiring larger datasets and resources.
>
> - "***SCAT and SNCAT is not clear***":
>
>   The main difference between SCAT and SNCAT is that SCAT has cross-attention module to learn interactions between objects while for SNCAT this cross-attention module is replaced by a similar capacity feedforward network. Additional clarifications are provided in the revised paper in Figures 1 and 2.
>
> - All typos, minor issues and missing placeholders for references are addressed and updated on the revised manuscript.

---

### Decision · Action_Editor_BmRB · 2025-11-13

**Recommendation:** Reject

**Audience:**

Yes

**Audience Explanation:**

Although the study would be of interest to the TMLR community, it would have more impact if:
- The system performance dependence on off-the-shelf segmentation models was reduced or handled.
- The slot formulation brittleness was addressed.
- The evaluation scope was expanded to more diverse datasets and additional SOTA baselines.

The study has the potential to be quite interesting to the TMLR community with the above changes.

**Claims And Evidence:**

No

**Claims Explanation:**

Overall, the revision was appreciated by the reviewers. The updates made the paper clearer and strengthened the experimental narrative.

The main claim of the paper is that modelling explicitly the motion of the main objects in a scene and their interaction allows for better video prediction without need for larger models or additional training data. This paper presents a study of the benefits of explicitly modeling individual object motions for video prediction within modern latent transformer frameworks. The goal is to assess whether object-level decomposition improves predictive quality and robustness compared to holistic, scene-level modeling. The paper includes evaluations on several benchmark datasets and compares against existing transformer-based video prediction methods, providing ablation studies and diagnostics to support its claims.

The reviewers found the claims in the paper to not be well supported due to the reliance on good performance from off-the-shelf segmentation models and the limited scope of the experiments. Using overly or under-segmented regions as inputs to downstream modules led to unstable performance, since segmentation errors can arise unpredictably across unseen images.

Reviewer QEQw made some suggestions for improvement in a future iteration that the authors are encouraged to consider: demonstrate that (1) the proposed method incorporates refinement mechanisms to handle potential segmentation errors, and to more clearly illustrate how it complements the outputs of an existing off-the-shelf model rather than merely depending on them; or (2) that even when the input segmentation is poor, the overall performance of the follow-up model remains largely unaffected.

The limited scope of the slot formulation was also a concern for the reviewers. The method depends on fixed class definitions and slot budgets, which limits its robustness when object counts vary or new classes appear. The evaluation scope was found to be too narrow: results are largely from small synthetic or constrained datasets, with limited comparison to recent slot-based or SOTA models.

Overall, the reviewers appreciated the care taken in conducting the reproducible study. With improved flexibility of the slot setting and more extensive empirical results, this paper could be a good contribution to TMLR.

**Resubmission Of Major Revision:**

The authors may consider submitting a major revision at a later time.